# Corrected Samplers for Discrete Flow Models

**Zhengyan Wan** [1]  **Yidong Ouyang** [2]  **Liyan Xie** [3]  **Hongyuan Zha** [† 4]  **Fang Fang** [† 1]  **Guang Cheng** [2]

## Abstract

Discrete flow models (DFMs) have been proposed to learn the data distribution on finite state space, offering a flexible framework as an alternative to discrete diffusion models. A line of recent work has studied samplers for discrete diffusion models, such as tau-leaping and Euler solver. However, these samplers require a large number of iterations to control discretization error, since the transition rates are frozen in time and evaluated at the initial state within each time interval. Moreover, theoretical results for these samplers often require boundedness conditions of the transition rate or they focus on a specific type of source distributions. To address those limitations, we establish non-asymptotic discretization error bounds for those samplers without any restriction on transition rates and source distributions, under the framework of discrete flow models. Furthermore, by analyzing a one-step lower bound of the Euler sampler, we propose two corrected samplers: *time-corrected sampler* and *location-corrected sampler*, which can reduce the discretization error of tau-leaping and Euler solver with almost no additional computational cost. We rigorously show that the location-corrected sampler has a lower complexity than existing parallel samplers. We validate the effectiveness of the proposed method by achieving better generation quality with reduced inference time on simulations and text-to-image generation tasks. Code can be found in https://github.com/WanZhengyan/Corrected-Samplers-for-Discrete-Flow-Models.

[1]School of Statistics, East China Normal University [2]Department of Statistics and Data Science, University of California, Los Angeles [3]Department of Industrial and Systems Engineering, University of Minnesota [4]School of Data Science, The Chinese University of Hong Kong, Shenzhen. Correspondence to: Fang Fang <ffang@sfs.ecnu.edu.cn>, Hongyuan Zha <zhahy@cuhk.edu.cn>.

*Proceedings of the 43$^{rd}$ International Conference on Machine Learning*, Seoul, South Korea. PMLR 306, 2026. Copyright 2026 by the author(s).

## 1. Introduction

Discrete flow models (DFMs) (Campbell et al., 2024; Gat et al., 2024; Shaul et al., 2025) arise in various practical applications, including but not limited to graph generation (Qin et al., 2025), visual generation and multimodal understanding (Wang et al., 2025), and video generation (Fuest et al., 2025; Deng et al., 2025). Unlike discrete diffusion models, which learn the time-reversal of a forward continuous-time Markov chain (CTMC), DFMs learn transition rates by matching conditional rate and estimated rate, parallel to continuous flow matching (Liu et al., 2023; Lipman et al., 2023; Albergo & Vanden-Eijnden, 2023). Despite their practical success (Wang et al., 2025), the theoretical analysis for DFMs remains limited. Existing efforts primarily focus on error analysis with the uniformization technique (Wan et al., 2025b) or on the estimation error of empirical risk minimization (Su et al., 2025). Although uniformization (Chen & Ying, 2024; Zhang et al., 2025; Ren et al., 2025a; Liang et al., 2025a) can simulate CTMCs in an exact way, it is inefficient due to the lack of parallelism. On the other hand, tau-leaping and Euler solvers require a large number of function evaluation to control discretization error (Ren et al., 2025a; Liang et al., 2025a;b). Additionally, Ren et al. (2025b) developed high-order tau-leaping (Hu et al., 2011) for discrete diffusion models and established theoretical results in an asymptotic regime; however, they do not provide iteration complexity in terms of dimension for the sampler.

In this paper, we rigorously establish the non-asymptotic total variation bound of tau-leaping and Euler sampler for discrete flow models in a unified framework. Rather than the conditions on the transition rate imposed in existing works, we only require the condition of finite estimation error. After analyzing a one-step lower bound of the Euler sampler, we propose two novel samplers: *time-corrected sampler* and *location-corrected sampler*. The time-corrected sampler corrects the time variable of the time schedule in the transition structure of the Euler sampler without additional computational cost. For the location-corrected sampler, we adjust the location of function evaluation after the first jump in each timestep (like a randomized midpoint sampler) by considering a (two-stage) jump process in each time interval, which only requires *at most two* function calls in each timestep. We also derive the non-asymptotic upper bounds for the proposed corrected samplers, and show that

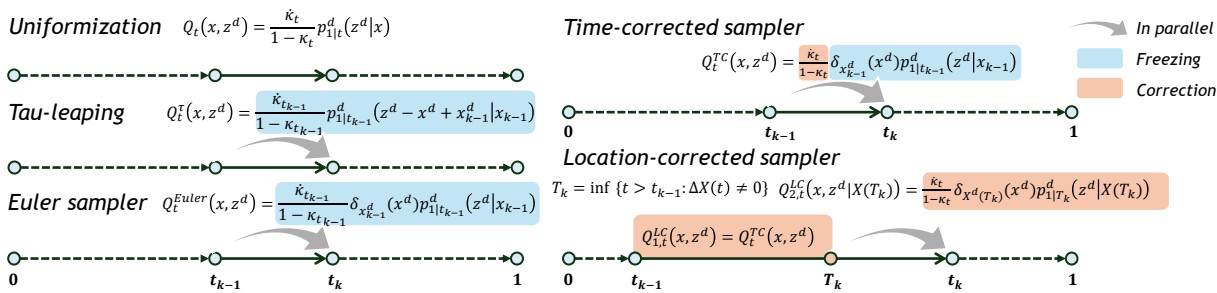

*Figure 1.* Overview of the transition rates of samplers in the $k$-th time interval ($z^d \neq x^d$). Tau-leaping and Euler sampler freeze the time variables of both time schedule and the posterior in the transition rates. For the time-corrected sampler, we correct the time variable of the time schedule without additional computational cost. For the location-corrected sampler, we not only correct the time variable of the schedule but also correct the location after the first jump in each timestep by constructing a two-stage transition rate; this sampler requires at most 2 function calls in each timestep.

the location-corrected sampler achieves smaller iteration complexity than other samplers; that is, it requires fewer steps to reach the same level of accuracy. See Figure 1 for the overview of those samplers. The proposed method achieves high generation quality with reduced inference time on simulation and text-to-image generation tasks.

**Contribution.** Our main contributions are as follows.

1. We establish the non-asymptotic upper bounds for tau-leaping and Euler samplers without imposing any constraints on the data/source distributions or boundedness conditions on transition rates, and further systematically analyze the one-step lower bound for the Euler sampler.

2. We propose two corrected samplers: *time-corrected sampler* and *location-corrected sampler*. The non-asymptotic bounds for the proposed corrected samplers are derived, which reveal that the location-corrected sampler achieves lower iteration complexity than tau-leaping and Euler samplers.

3. We evaluate the proposed samplers on the low-dimensional simulation and high-dimensional text-to-image generation tasks, demonstrating that the corrected samplers achieve competitive performance compared to other parallel samplers.

All technical proofs, notations, and some further detailed discussions are deferred to the Appendix.

## 2. Discrete Flow-Based Models and Continuous-Time Jump Processes

In this section, we briefly review background on discrete flow-based models and introduce a class of two-stage jump processes for constructing samplers with history-dependent transition rates.

### 2.1. Discrete Flow Models

We consider a target data distribution with discrete support $\mathcal{S}^{\mathcal{D}}$, where $\mathcal{S} = \{1, 2, \ldots, |\mathcal{S}|\}$. Let $p_1(x)$ denote its probability mass function. Our goal is to construct a flow-based model that produces samples from this distribution. To this end, we first define continuous-time Markov chains (CTMCs, Norris, 1998) on the probability space $(\Omega, \mathcal{F}, \mathbb{P})$ as follows.

**Definition 1** (CTMC). *Let $Q_t(x, z)_{x,z \in \mathcal{S}^{\mathcal{D}}}$ be a time-varying rate matrix satisfying: (a) $Q_t(x, z)$ is continuous in $t$; (b) $Q_t(x, z) \geq 0$ for any $z \neq x$; (c) $\sum_{z \in \mathcal{S}^{\mathcal{D}}} Q_t(x, z) = 0$ for any $x \in \mathcal{S}^{\mathcal{D}}$. A stochastic process $\{X(t)\}_{t \geq 0}$ taking values in $\mathcal{S}^{\mathcal{D}}$ is called a continuous time Markov chain with transition rate matrix $Q_t(x, z)_{z,x \in \mathcal{S}^{\mathcal{D}}}$ if for any $z, x \in \mathcal{S}^{\mathcal{D}}$ and $h > 0$, we have*

$$\mathbb{P}(X(t+h) = z | X(t) = x) = \delta_x(z) + Q_t(x, z)h + o(h),$$

*and*

$$\mathbb{P}(X(t+h) = z | \mathcal{F}_t) = \mathbb{P}(X(t+h) = z | X(t)),$$

*where $\mathcal{F}_t = \sigma(\{X(s) : 0 \leq s \leq t\})$ denotes the natural filtration.*

Given a CTMC $\{X(t)\}_{t \geq 0}$ with transition rate $Q_t$, its marginal densities $\{p_t\}_{t \geq 0}$ satisfy the following Kolmogorov forward equation:

$$\dot{p}_t(x) = \underbrace{\sum_{z \neq x} p_t(z) Q_t(z, x)}_{\text{incoming flux}} - \underbrace{\sum_{z \neq x} p_t(x) Q_t(x, z)}_{\text{outgoing flux}}.$$

In this work, we say $Q_t$ can generate the probability path $p_t$, if they satisfy the above Kolmogorov forward equation.

We aim to learn a transition rate $Q_t$ of a CTMC $\{X(t)\}_{0 \leq t \leq 1}$ that can transport from a source distribution $p_0$ to a target data distribution $p_1$. To obtain such a transition rate for sampling, a natural method is to learn the conditional

expectation of the conditional transition rate $Q_t(x, z|x_1)$ that generates the conditional probability path $p_{t|1}(\cdot|x_1)$, since $Q_t(x, z) = \mathbb{E}[Q_{\mathbf{t}}(x, z|X(1))|X(\mathbf{t}) = x, \mathbf{t} = t]$ can generate the target probability path $p_t$ (see Proposition 3.1 of Campbell et al., 2024), i.e., it satisfies the Kolmogorov forward equation, where $X(\mathbf{t}) \sim p_{\mathbf{t}|1}(\cdot|X(1))$ given $X(1)$ and $\mathbf{t} \sim \mathcal{U}([0, 1])$. Therefore, it suffices to specify the conditional probability path and conditional transition rate.

It is worth noting that in the sampling stage, considering a $|\mathcal{S}|^{\mathcal{D}}$-dimensional vector-valued function $(Q_t(x, z))_{z \in \mathcal{S}^{\mathcal{D}}}$ of current time $t$ and state $x$ is intractable when $\mathcal{D}$ is relatively large. To handle such a high-dimensional scenario, a common approach is to construct a coordinate-wise conditional probability path and transition rate as follows:

$$
\begin{aligned}
p_{t|1}(x|x_1) &= \prod_{d=1}^{\mathcal{D}} p_{t|1}^d(x^d|x_1^d); \\
Q_t(x, z|x_1) &= \sum_{d=1}^{\mathcal{D}} \delta_{x^{\backslash d}}(z^{\backslash d}) Q_t^d(x^d, z^d|x_1^d),
\end{aligned} \tag{1}
$$

which means that the elements of the vector $X(t)$ are independent conditional on $X(1)$. Here, $Q_t^d(x^d, z^d|x_1^d)$ is the conditional transition rate that generates the conditional probability path $p_{t|1}^d$; then $Q_t(x, z|x_1)$ can generate $p_{t|1}$ by the following proposition.

**Proposition 1** (Independent coupling). *If the marginal transition rate $Q_t^d(x^d, z^d)$ can generate marginal probability path $p_t^d(x^d)$ for any $d \in [\mathcal{D}]$, then the joint transition rate $Q_t(x, z) = \sum_{d=1}^{\mathcal{D}} \delta_{x^{\backslash d}}(z^{\backslash d}) Q_t^d(x^d, z^d)$ can generate $p_t(x_t) = \prod_{d=1}^{\mathcal{D}} p_t^d(x_t^d)$.*

**Remark 1** (Parallel samplers). *Proposition 1 is useful for constructing parallel samplers. To be specific, if the transition rate of a sampler has the form of $Q_t(x, z; x_{k-1}) = \sum_{d=1}^{\mathcal{D}} \delta_{x^{\backslash d}}(z^{\backslash d}) Q_t^d(x^d, z^d; x_{k-1})$ in the time interval $[t_{k-1}, t_k]$, then each token can be updated independently via Proposition 1, where $x_{k-1}$ is the initial state in this interval (e.g., transition rate (equation 6) of tau-leaping).*

In this work, we focus mainly on the widely used probability path and the associated conditional transition rate from prior work (Campbell et al., 2024; Gat et al., 2024):

$$
\begin{aligned}
p_{t|1}^d(x^d|x_1^d) &= (1 - \kappa_t) p_0^d(x^d) + \kappa_t \delta_{x_1^d}(x^d); \\
Q_t^d(x^d, z^d|x_1^d) &= \frac{\dot{\kappa}_t}{1 - \kappa_t} (\delta_{x_1^d}(z^d) - \delta_{x^d}(z^d)),
\end{aligned} \tag{2}
$$

where $\kappa_t : [0, 1] \to [0, 1]$ is a non-decreasing and continuous function satisfying $\kappa_0 = 0$ and $\kappa_1 = 1$. Note that the conditional transition rate will blow up as $t \to 1^-$. Thus, in the sampling stage, we employ the early stopping technique; that is, we only consider the time interval $[0, 1 - \delta]$ for a sufficiently small positive parameter $\delta$.

After defining the conditional path and rate, the (oracle) unconditional transition rate is given by

$$
\begin{aligned}
Q_t(x, z) &= \sum_{d=1}^{\mathcal{D}} \delta_{x^{\backslash d}}(z^{\backslash d}) \sum_{x_1^d} Q_t^d(x^d, z^d|x_1^d) p_{1|t}^d(x_1^d|x) \\
&\triangleq \sum_{d=1}^{\mathcal{D}} \delta_{x^{\backslash d}}(z^{\backslash d}) Q_t^d(x, z^d),
\end{aligned} \tag{3}
$$

where $p_{1|t}^d(x_1^d|x) = \sum_{x_1^{\backslash d}} p_{1|t}(x_1|x)$. Therefore, it suffices to consider a sparse rate matrix $Q_t$ satisfying $Q_t(x, z) = 0$ for any $z, x \in \mathcal{S}^{\mathcal{D}}$ with Hamming distance $d^H(z, x) > 1$. In our analysis, unlike Chen & Ying (2024); Zhang et al. (2025); Pham et al. (2025); Wan et al. (2025b), we do not restrict the source distribution $p_0$ to be of a uniform distribution.

**Training.** To learn the transition rate, a common approach is to minimize the Bregman divergence between the conditional generator and the estimated generator (Holderrieth et al., 2025; Lipman et al., 2024; Shaul et al., 2025; Wan et al., 2025b):

$$
\mathbb{E}\Big[ \sum_{z \neq X(\mathbf{t})} D_F\Big(Q_{\mathbf{t}}(X(\mathbf{t}), z|X(1)) \Big\| Q_{\mathbf{t}}(X(\mathbf{t}), z)\Big)\Big], \tag{4}
$$

where $D_F$ is the Bregman divergence induced by $F(x) = x \log x$. In practice, one can train a logits model by minimizing a training objective in terms of posterior (Equation (16) in Appendix), which is *equal* to the above formula.

### 2.2. Two-Stage Jump Processes

In this subsection, we introduce a class of jump processes, which includes CTMCs as a special case. Such a broader class allows transition rates to depend on history information (e.g. a stopping time), enabling refinement of samplers.

In general, we can consider the following (two-stage) transition rate:

$$
\begin{aligned}
Q_t(x, z) = {}&\mathbb{1}(T^X \geq t) \underbrace{Q_{1,t}(x, z)}_{\text{first-stage rate}} \\
&+ \mathbb{1}(T^X < t) \underbrace{Q_{2,t}(x, z; X(T^X))}_{\text{second-stage rate}},
\end{aligned} \tag{5}
$$

where $T^X = \inf\{t > 0 : \Delta X(t) \neq 0\}$ is the exit time of the process $X(t)$, $Q_{1,t}$ is a deterministic transition rate, and $Q_{2,t}$ is a transition rate depending on the location $X(T^X)$ after the first jump. Both transition rates $Q_{1,t}$ and $Q_{2,t}$ satisfy the rate properties defined in Definition 1 given $X(T^X)$. In particular, $Q_t(x, z)$ is a transition rate for CTMC if $Q_{2,t}(x, z; X(T^X)) = Q_{1,t}(x, z)$.

Similar to CTMCs, we can also derive several theoretical results for two-stage jump processes, including uniformization and a Girsanov-type theorem; see Section C for details.

**Exit time.** Conditioning on the initial point $X(0)$, one can directly calculate the distribution of the exit time of the jump process $X(t)$, which is crucial for our analysis and for deriving algorithms with different transition structures.

**Proposition 2** (Exit time). *Under the assumptions in Proposition 3, assume the distribution of $X(0)$ is a point mass at $x$ and define the exit time $T^X = \min\{t > 0 : \Delta X(t) \neq 0\}$. Denote $\mathcal{Q}(x, t) = \int_0^t Q_{1,s}(x, x)ds$. Then we have*

$$\mathbb{P}(T^X > t) = \exp\left(\mathcal{Q}(x, t)\right).$$

For a small time interval $[0, \tau]$, by Proposition 2, $X(t)$ has at least one jump in this interval with probability $1 - \exp\left(\mathcal{Q}(x, t)\right)$ and there is no jump in this interval with probability $\exp\left(\mathcal{Q}(x, t)\right)$.

## 3. Tau-Leaping and Euler Samplers

In this section, we discuss the tau-leaping algorithm and Euler sampler, and derive non-asymptotic upper bounds for those samplers. Finally, we will establish a one-step lower bound for Euler sampler, which motivates us to construct corrected samplers.

In the sequel, let $Q_t$ be the oracle transition rate defined in equation 3. Consider the time discretization scheme: $0 = t_0 < t_1 < \cdots < t_K = 1 - \delta$, where $\delta$ is the early stopping parameter. On the $k$-th time interval $[t_{k-1}, t_k]$, the tau-leaping process is a CTMC with transition rate $Q_t^\tau(x, z) = Q_{t_{k-1}}(x_{k-1}, z - x + x_{k-1})$, which not only freezes the time variable $t = t_{k-1}$ but also fixes the first state argument of the transition rate at $x_{k-1}$ such that only one function call is needed in this interval (Campbell et al., 2022). Here $x_{k-1}$ is a fixed initial point at time $t_{k-1}$. Additionally, by Proposition 1 and noting that

$$Q_t^\tau(x, z) = \sum_{d=1}^{\mathcal{D}} \delta_{x^{\backslash d}}(z^{\backslash d}) Q_{t_{k-1}}^d(x_{k-1}, z^d - x^d + x_{k-1}^d), \tag{6}$$

the coordinates of the tau-leaping process are independent, which make it possible to achieve parallelism. The tau-leaping algorithm is presented in Algorithm 2.

However, as discussed in (Campbell et al., 2022), the tau-leaping process with rate $Q_t^\tau$ is intractable if the state space is restricted in $\mathcal{S}^{\mathcal{D}}$ rather than $\mathbb{Z}^{\mathcal{D}}$, which leads to an infinite KL divergence between the path measure of the idealized process (e.g. exact simulation by uniformization) and that of tau-leaping process. A natural method is to restrict the number of jumps of tau-leaping algorithm for each dimension (e.g. Algorithm 1 in Campbell et al. (2022) and Algorithm 3 in Liang et al. (2025b)) and use the following transition rate:

$$Q_t^{\text{Euler}}(x, z) = \sum_{d=1}^{\mathcal{D}} \delta_{x^{\backslash d}}(z^{\backslash d}) \delta_{x_{k-1}^d}(x^d) Q_{t_{k-1}}^d(x_{k-1}, z^d), \tag{7}$$

which leads to an algorithm similar to the always-valid Euler solver used in (Shaul et al., 2025; Wang et al., 2025). Since there is at most one jump in each dimension, by Proposition 2, we can sample from Bernoulli distribution with probability $\exp((t_{k-1} - t_k)\frac{\dot{\kappa}_{t_{k-1}}}{1-\kappa_{t_{k-1}}}\lambda_k^d)$ to decide whether there is a jump for the $d$-th dimension in the $k$-th time interval, where $\lambda_k^d = \sum_{z^d \neq x_{k-1}^d} p_{1|t_{k-1}}^d(z^d|x_{k-1})$. This Euler solver (or so-called truncated tau-leaping) is described in Algorithm 3.

**Remark 2** (Masked source distribution). *By Proposition 6 in (Gat et al., 2024), if the source distribution is a point mass at $\mathtt{m}$, then the posterior is time-independent. Thus, if there is no jump in the $k$-th time interval, then no function evaluation is required in the $(k+1)$-th time interval. In this case, for any $K \in \mathbb{N}$, the number of function evaluations is not greater than $\mathcal{D}$.*

### 3.1. Non-Asymptotic Error Bound

In this subsection, we present the non-asymptotic total variation convergence results for Tau-leaping algorithm and Euler solver. The non-asymptotic upper bound identifies the sources of discretization error, which are later shown to match the one-step lower bound. We highlight that our results hold for any source distribution and data distribution without any additional assumption for the oracle transition rate and neural network class. Throughout this article, we denote $M_k = \sup_{t \in [t_{k-1}, t_k]} |\dot{\kappa}_t/(1 - \kappa_t)|$, which is the upper bound of the conditional transition rate $Q_t(x, z|x_1)$ for $\mathrm{d}^H(x, z) = 1$ in the $k$-th time interval.

**Remark 3** (Target set mismatch for tau-leaping). *The reason why we only consider the total variation as metric here is that the target set of tau-leaping process and idealized process mismatch; that is, $Q_t^\tau(X(t-), z)$ and $Q_t(X(t-), z)$ might have different supports as a function of $z$ if there are multiple jumps in the time intervals, which results in an infinite KL divergence by Theorem 5. Unlike KL divergence, the total variation has the trivial bound 1.*

**Assumption 1** (Estimation error).

$$\sum_{k=1}^{K} \mathbb{E}\left[\mathbb{E}_{\mathbb{P}_k^{Alg}}\left(\int_{t_{k-1}}^{t_k}\left\{\sum_{z \neq X_k^{Alg}(s)} D_F\left(Q_s^{Alg}||\hat{Q}_s^{Alg}\right)(X_k^{Alg}(s), z)\right\}ds\right)\right]$$
$$= 2(\varepsilon_{Est}^{Alg})^2 < \infty,$$

*where $\hat{Q}$ is the estimated rate depending on a dataset $\mathbb{D}_n$, $X_k^{Alg}(t)$ is a process independent of $\mathbb{D}_n$ with the rate $Q_t^{Alg}$ and $\mathbb{P}_k^{Alg}$ is the path measure of $X_k^{Alg}(t)$ in*

$[t_{k-1}, t_k]$. *Notably, this term is the KL divergence between two path measures of the processes with $Q_t^{Alg}$ and $\hat{Q}_t^{Alg}$. In particular, for tau-leaping and Euler sampler, the estimation error can be bounded by $2(\varepsilon_{Est}^{Alg})^2 \leq \mathcal{D}|\mathcal{S}| \sum_{k=1}^{K} \tau_k \sup_{d^H(x,z)=1} \mathbb{E}_{\mathbb{D}_n}[D_F(Q_{t_{k-1}}^{Alg}||\hat{Q}_{t_{k-1}}^{Alg})(x,z)]$.*

**Theorem 1** (Tau-leaping and Euler sampler). *Assume that $\max_{k\in[K]}\{\tau_k \mathcal{D} M_k\} \leq 1$. The tau-leaping and Euler sampler have the following bound for the total variation between the estimated and the oracle marginal distribution at time $t = 1 - \delta$ (up to a logarithmic factor):*

$$\mathbb{E}[TV(p_{1-\delta}, \hat{p}_{1-\delta}^{Alg})]$$
$$\leq \sqrt{\frac{5}{2}\mathcal{D}|\mathcal{S}|\sum_{k=1}^{K}\tau_k^2 H_k} + \sqrt{3\mathcal{D}^2|\mathcal{S}|\sum_{k=1}^{K}\tau_k^2 M_k^2} + \varepsilon_{Est}^{Alg},$$

*where $H_k = \sup_{t\in[t_{k-1},t_k]}|\frac{\ddot{\kappa}_t(1-\kappa_t)+(\dot{\kappa}_t)^2}{(1-\kappa_t)^2}|$, and $\varepsilon_{Est}^{Alg}$ is defined in Assumption 1.*

### 3.2. One-Step Analysis

To better understand the discretization error in Theorem 1, we conduct a non-asymptotic one-step analysis for the Euler solver, and give the lower bound for the KL divergence between the path measure of the Euler solver and that of the idealized process. This error bound provides insights for us to construct efficient samplers that reduce the discretization error of Euler sampler.

**Theorem 2** (One-step lower bound for Euler sampler). *Given the initial point $x_{k-1}$, let $\mathbb{P}_k$ and $\mathbb{P}_k^{Euler}$ be the path measures of the processes with transition rates $Q_t$ and $Q_t^{Euler}$ in $[t_{k-1}, t_k]$, respectively. Assume that $Q_t(x,z) > 0$ for all $x, z \in \mathcal{S}^{\mathcal{D}}$ with $d^H(x,z) = 1$ and $t \in [t_{k-1}, t_k]$. If $\tau_k Q_{t_{k-1}}(x_{k-1}, x_{k-1}) \geq -\log 2$, then we have*

$$D_{KL}(\mathbb{P}_k^{Euler}||\mathbb{P}_k) \geq \frac{1}{4}\tau_k \mathcal{D}|\mathcal{S}|L_k' + \frac{1}{32}\mathcal{D}^2|\mathcal{S}|^2\tau_k^2\eta_k, \quad (8)$$

*where*

$$L_k' = \inf_{d^H(x,z)=1}\int_{t_{k-1}}^{t_k}\left\{\frac{1}{\tau_k}D_F(Q_{t_{k-1}}(x,z)||Q_s(x,z))\right\}ds;$$
$$\eta_k = \inf\left\{Q_{t_{k-1}}^d(x_{k-1},x^d)D_F(Q_{t_{k-1}}^{d'}(x_{k-1},z^{d'})||Q_s^{d'}(x,z^{d'}))\right\}.$$

*Here, the second infimum is taken over $d' \neq d \in [\mathcal{D}]$, $z^{d'} \neq x^{d'}, x^d \neq x_{k-1}^d, d^H(x, x_{k-1}) = 1$ and $s \in [t_{k-1}, t_k]$.*

The first term of the RHS in equation 8 depends on the time-Lipchitz $L_k'$ of the transition rate, which is related the first term of the upper bound in Theorem 1 when the posterior $p_{1|t}$ is time-independent. We will propose a time-corrected sampler with a time-varying transition rate in Section 4 to reduce this error. Additionally, the second term of the RHS in equation 8 is related to the event that there is at least one

jump in the $k$-th time interval, which is related to the second term of the upper bound in Theorem 1. We will propose a location-corrected sampler in Section 5 to control this error.

## 4. Time-Corrected Sampler

As discussed in the previous section, the transition rates for tau-leaping and Euler sampler are time-homogeneous rates, which not only freeze the time variable of the posterior $p_{1|t}$ but also freeze that of the time schedule $\kappa_t$. In this section, we aim to propose a new sampler with time-varying rate that can reduce the error from the time-Lipschitz constant of the transition rate for the Euler sampler.

By equation 2 and equation 3, the oracle transition rate $Q_t$ can be written as the posterior $p_{1|t}$ weighted by a term related to the time schedule: $Q_t^d(x, z^d) = \frac{\dot{\kappa}_t}{1-\kappa_t}p_{1|t}^d(z^d|x)$, where $z^d \neq x^d$. In general, calculating the exact distribution of the exit time with a time-dependent posterior is intractable, since its survival function is related to the time integral of the transition rate (Proposition 2). To mitigate this issue, we only freeze the time variable of the posterior and consider the following time-corrected transition rate in $t \in [t_{k-1}, t_k]$, which also allows us to update all tokens in parallel:

$$Q_t^{\text{TC}}(x, z)$$
$$= \frac{\dot{\kappa}_t}{1-\kappa_t}\sum_{d=1}^{\mathcal{D}}\delta_{x^{\backslash d}}(z^{\backslash d})\delta_{x_{k-1}^d}(x^d)p_{1|t_{k-1}}^d(z^d|x_{k-1}). \quad (9)$$

Intuitively, equation 9 keeps the posterior $p_{1|t}$ evaluated at the beginning of the interval, so the model is evaluated only once in each timestep. At the same time, the schedule factor $\dot{\kappa}_t/(1-\kappa_t)$ remains time-varying, which leads to the closed-form survival function in equation 10.

### 4.1. Algorithm

Let $\lambda_k^d = \sum_{z^d \neq x_{k-1}^d}p_{1|t_{k-1}}^d(z^d|x_{k-1})$. By Proposition 1 and Proposition 2, we can sample exit times in parallel for all dimensions from a distribution with the survival function

$$\mathbb{P}(T_k^d > t) = \exp\left(-\lambda_k^d\int_{t_{k-1}}^{t}\frac{\dot{\kappa}_s}{1-\kappa_s}ds\right) = \left(\frac{1-\kappa_t}{1-\kappa_{t_{k-1}}}\right)^{\lambda_k^d} \quad (10)$$

at the $k$-th timestep to decide whether there is a jump for the $d$-th dimension; that is, if $T_k^d \geq t_k$, then we change $x_{k-1}^d$ to $z^d \neq x_{k-1}$ with probability $p_{1|t_{k-1}}(z^d|x_{k-1})/\lambda_k^d$. Similar to Euler sampler, instead of sampling $T_k^d$, we can directly sample Bernoulli distribution with probability $\left(\frac{1-\kappa_{t_k}}{1-\kappa_{t_{k-1}}}\right)^{\lambda_k^d}$; the details can be found in Algorithm 4.

## 4.2. Non-Asymptotic Error Bound

In this subsection, we establish the non-asymptotic error bound for proposed time-corrected sampler.

**Theorem 3** (Time-corrected sampler). *Assume that* $\max_{k\in[K]}\{\tau_k\mathcal{D}M_k\}\leq 1$. *The time-corrected sampler has the following bound for the total variation between the estimated and the oracle marginal distribution at time* $t = 1-\delta$ *(up to a logarithmic factor):*

$$\mathbb{E}[TV(p_{1-\delta},\hat{p}_{1-\delta}^{TC})] \leq \sqrt{2\mathcal{D}^2|\mathcal{S}|\sum_{k=1}^{K}\tau_k^2 M_k^2} + \varepsilon_{Est}^{TC},$$

*where* $\varepsilon_{Est}^{TC}$ *is defined in Assumption 1.*

Compared to the error bound of the tau-leaping and Euler sampler in Theorem 1, the non-asymptotic error bound of the proposed time-corrected sampler reduces the error from freezing the time schedule in each timestep (first term of the error bound in Theorem 1).

# 5. Location-Corrected Sampler

In the previous section, we reduce the error from the time-Lipschitz of the transition rate we used in our sampling procedure. In this section, we tackle the error from the event that there is at least one jump in each timestep. A natural approach is to adjust the location after the first jump in each time interval. Unlike prior samplers, we have to introduce the information of the first exit time on each time interval in our transition structure. We consider the following rates in the two-stage transition rate (equation 5) for the $k$-th time interval:

$$Q_{1,t}^{\mathrm{LC}}(x,z) = \frac{\dot{\kappa}_t}{1-\kappa_t}\sum_{d=1}^{\mathcal{D}}\delta_{x^{\backslash d}}(z^{\backslash d})p_{1|t_{k-1}}^d(z^d|x);$$

$$Q_{2,t}^{\mathrm{LC}}(x,z|X(T_k)) = \frac{\dot{\kappa}_t}{1-\kappa_t}\sum_{d=1}^{\mathcal{D}}\Big[\delta_{x^{\backslash d}}(z^{\backslash d})\delta_{X^d(T_k)}(x^d) \\ \times p_{1|T_k}^d(z^d|X(T_k))\Big].$$

(11)

where $T_k > t_{k-1}$ is the exit time of the jump process.

Intuitively, the first-stage rate $Q_{1,t}^{\mathrm{LC}}$ uses the initial posterior at $t_{k-1}$ before the first jump. Once a jump occurs at $T_k$, the second-stage rate $Q_{2,t}^{\mathrm{LC}}$ reevaluates the posterior at the new location $X(T_k)$ and the current time $T_k$, thereby correcting the location-dependent error suggested by the one-step lower bound in Theorem 2.

## 5.1. Algorithm

Let $Q_t^{\mathrm{LC}}$ be the (two-stage) location-corrected transition rate with the choice in equation 11. By uniformization

construction, similar to equation 10, we first sample the exit time $T_k$ from a distribution with survival function

$$\mathbb{P}(T_k > t) = \left(\frac{1-\kappa_t}{1-\kappa_{t_{k-1}}}\right)^{\lambda_k},\qquad(12)$$

where $\lambda_k = \sum_{d=1}^{\mathcal{D}}\sum_{z^d\neq x_{k-1}^d}p_{1|t_{k-1}}^d(z^d|x_{k-1})$. If $T_k \geq t_k$, then we keep the current state and enter the next time interval. If $T_k < t_k$, we jump to the new location $X_k^{\mathrm{LC}}(T_k) = z$ with probability

$$\frac{\sum_{d=1}^{\mathcal{D}}\delta_{x_{k-1}^{\backslash d}}(z^{\backslash d})p_{1|t_{k-1}}^d(z^d|x_{k-1})}{\lambda_k}$$

in the first stage; then we reevaluate the posterior model using the current time $T_k$ and the new state $X_k^{\mathrm{LC}}(T_k)$ and follow the same parallel procedure of the time-corrected sampler in the interval $[T_k, t_k]$. In practice, if $\kappa_t$ is a strictly increasing function, then we can sample $T_k$ by generating a random variable of exponential distribution (Lemma 5). The location-corrected algorithm can be found in Algorithm 5.

**Remark 4** (Randomized midpoint sampler). *Our proposed location-corrected sampler has a randomized midpoint under the event* $\{T_k < t_k\}$ *in the* $k$-th time interval. Compared to the samplers introduced in Hu et al. (2011); Ren et al. (2025b); Monsefi et al. (2025) (which requires a deterministic number of function calls in each time interval), our sampler has fewer function calls with the same time-discretization scheme, since* $\{T_k \geq t_k\}$ *holds with high probability when* $\tau_k$ *is sufficiently small.*

## 5.2. Non-Asymptotic Error Bound

In this subsection, we present the non-asymptotic error bound for the (two-stage) location-corrected sampler.

**Theorem 4** (Location-corrected sampler). *Assume that* $\max_{k\in[K]}\{\tau_k\mathcal{D}M_k\}\leq \log 2$. *The location-corrected sampler has the following bound for the total variation between the estimated and the oracle marginal distribution at time* $t = 1-\delta$ *(up to a logarithmic factor):*

$$\mathbb{E}[TV(p_{1-\delta},\hat{p}_{1-\delta}^{LC})]$$
$$\leq \sqrt{2\mathcal{D}^3|\mathcal{S}|\sum_{k=1}^{K}\tau_k^3 M_k^3\{(L_k^p/(\tau_k\mathcal{D}^2 M_k^2))\vee 1\}} + \varepsilon_{Est}^{LC},$$

*where*

$$L_k^p = \sup_{t\in[t_{k-1},t_k],d\in[\mathcal{D}],x\in\mathcal{S}^{\mathcal{D}},z^d\in\mathcal{S}\backslash\{x^d\}}|\dot{p}_{1|t}^d(z^d|x)|,$$

*and* $\varepsilon_{Est}^{LC}$ *is defined in Assumption 1.*

The convergence rate of the location-corrected sampler depends on the time-Lipschitz constant of the posterior

*Table 1.* Summary of the iteration complexity of samplers with the linear schedule; that is, the number of steps required to guarantee a total variation error $\mathbb{E}[TV(p_{1-\delta}, \hat{p}_{1-\delta}^{\text{Alg}})]$ at most $2\varepsilon_{\text{Est}}^{\text{Alg}}$ (up to a logarithmic factor). The iteration complexity is decomposed into four quantities: the dimension $\mathcal{D}$, the vocabulary size $|\mathcal{S}|$, the early-stopping parameter $\delta$, and the estimation error $\varepsilon_{\text{Est}}^{\text{Alg}}$, where $\mathcal{D}$ is the quantity of primary interest. The complexity of the location-corrected sampler depends on the time-Lipschitz regularity of the posterior; if posterior is time-independent, the dependence on $\mathcal{D}$ can be improved from $\mathcal{D}^2$ to $\mathcal{D}^{3/2}$.

| Samplers | Complexity |
|---|---|
| Tau-leaping and Euler sampler (Theorem 1) | $\mathcal{O}\left(\frac{\mathcal{D}^2\|\mathcal{S}\|\log^2(\frac{1}{\delta})}{(\epsilon_{\text{Est}}^{\text{Alg}})^2}\right)$ |
| Time-corrected sampler (Theorem 3) | $\mathcal{O}\left(\frac{\mathcal{D}^2\|\mathcal{S}\|\log^2(\frac{1}{\delta})}{(\epsilon_{\text{Est}}^{\text{Alg}})^2}\right)$ |
| (Two-stage) Location-corrected sampler (Theorem 4) | $\mathcal{O}\left(\frac{\mathcal{D}\|\mathcal{S}\|\log^2(\frac{1}{\delta})\max_{k=1}^{K}\{L_k^p/M_k\}}{(\epsilon_{\text{Est}}^{\text{Alg}})^2} \vee \frac{\mathcal{D}^{\frac{3}{2}}\|\mathcal{S}\|^{\frac{1}{2}}\log^{\frac{3}{2}}(\frac{1}{\delta})}{\epsilon_{\text{Est}}^{\text{Alg}}}\right)$ |

$p_{1|t}$ case-by-case. If the posterior is time-independent (e.g. one chooses a masked source distribution and mixture path), then the total variation error bound reduces to $\sqrt{2\mathcal{D}^3|\mathcal{S}|\sum_{k=1}^{K}\tau_k^3 M_k^3} + \varepsilon_{\text{Est}}^{\text{LC}}$. Moreover, the time-Lipschitz constant also relies on the dependence of data distribution. For instance, if $p_1(x) = \prod_{d=1}^{\mathcal{D}} p_1^d(x^d)$, then $L_k^p \leq M_k$ by Kolmogorov backward equation (Lemma 2); in this case, the error bound is $\sqrt{2\mathcal{D}|\mathcal{S}|\sum_{k=1}^{K}\tau_k^2 M_k^2} + \varepsilon_{\text{Est}}^{\text{LC}}$. In the worst case, $L_k^p \leq \mathcal{D}M_k$, and the error bound is $\sqrt{2\mathcal{D}^2|\mathcal{S}|\sum_{k=1}^{K}\tau_k^2 M_k^2} + \varepsilon_{\text{Est}}^{\text{LC}}$. With a lower-boundedness condition for the posterior $p_{1|t}$, we can obtain an elegant upper bound which does not depend on $L_k^p$; see Section D.2 for details.

### 5.3. Choosing Time-Discretization Scheme

In this subsection, we aim to find a suitable time-discretization scheme $\{t_k^*\}_{k\in[K]}$ by optimizing the discretization error bound. Notice that the error bounds established in Theorems 3 and 4 have the following form (we ignore the logarithmic factor and view $L_k^p = 0$ or $\asymp \mathcal{D}M_k$):

$$\mathbb{E}[TV(p_{1-\delta}, \hat{p}_{1-\delta}^{\text{Alg}})] \leq \underbrace{\sqrt{2\mathcal{D}^{J+1}|\mathcal{S}|\sum_{k=1}^{K}\tau_k^{J+1} M_k^{J+1}}}_{\text{discretization error}} + \varepsilon_{\text{Est}}^{\text{Alg}},$$

(13)

where $J \in \{1, 2\}$. By Jensen's inequality, the discretization error bound achieves the minimum at $\{\tau_k^*\}_{k\in[K]}$ that satisfies $\sum_{k=1}^{K}\tau_k^* = 1 - \delta$ and $\tau_k^* M_k = c^*$ for all $k \in [K]$.

**Optimal scheme for linear schedule.** Consider the linear schedule $\kappa_t = t$. In this case, $M_k = 1/(1-t_k^*) = 1/(\delta + \sum_{i=k+1}^{K}\tau_i^*)$. Thus, we have $\tau_K^* = \delta c^*$ and $\sum_{i=k}^{K}\tau_i^* = \delta c^* + (c^* + 1)\sum_{i=k+1}^{K}\tau_i^*$. By induction, we get $1 - \delta = \sum_{i=k}^{K}\tau_k^* = \delta c^* \sum_{k=1}^{K}(c^* + 1)^{k-1} = \delta\{(c^* + 1)^K - 1\}$, which implies that $c^* = \delta^{-1/K} - 1$. Consequently, the

time-discretization scheme for linear schedule is

$$\tau_k^* = \delta^{\frac{k-1}{K}} - \delta^{\frac{k}{K}}.$$

(14)

**Iteration complexity for linear schedule.** If we choose the time-discretization scheme in equation 14 for the linear schedule, then the discretization error in equation 13 can be bounded by $\sqrt{2\mathcal{D}^{J+1}|\mathcal{S}|K(\frac{2}{K}\log\frac{1}{\delta})^{J+1}}$ provided that $\frac{1}{K}\log\frac{1}{\delta} \leq \log 2$. Table 1 summarizes the iteration complexity of each sampler (up to a logarithmic factor); that is, the number of steps required to guarantee a total variation at most $2\varepsilon_{\text{Est}}^{\text{Alg}}$.

**Remark 5** (Multistage location-corrected sampler). *Similar to the two-stage location-corrected sampler, one can construct a $J$-stage location-corrected sampler, which can be viewed as an $J$-order approximation of uniformization algorithm if the posterior is time-independent. In this scenario, the total variation bound would be $\sqrt{2\mathcal{D}^{J+1}|\mathcal{S}|\sum_{k=1}^{K}\tau_k^{J+1} M_k^{J+1}} + \varepsilon_{\text{Est}}^{\text{LC}}$ using similar arguments in the proof of Theorem 4; the discretization error would go to zero as $J \to \infty$ for fixed $K \in \mathbb{Z}^+$. In this case, the iteration complexity is given by $\mathcal{O}\left(\mathcal{D}^{\frac{J+1}{J}}|\mathcal{S}|^{\frac{1}{J}}\log^{\frac{J+1}{J}}(\frac{1}{\delta})/(\varepsilon_{\text{Est}}^{\text{Alg}})^{\frac{2}{J}}\right)$, which is linear in dimension $\mathcal{D}$ for sufficiently large $J$ (say, $J \asymp \mathcal{D}$). However, if the posterior is time-dependent, although a multistage sampler is used, the error from the time-Lipschitz constant of the posterior might dominate the discretization error.*

## 6. Experiments

In this section, we evaluate our proposed corrected samplers on low-dimensional dataset and text-to-image generation task. Additional experiments and implementation details can be found in Section H.

### 6.1. Low-Dimensional Simulations

We conduct simulations on low-dimensional data set with $\mathcal{D} = 9$ and $|\mathcal{S}| = 8$ (the size of the state space is $8^9$). In

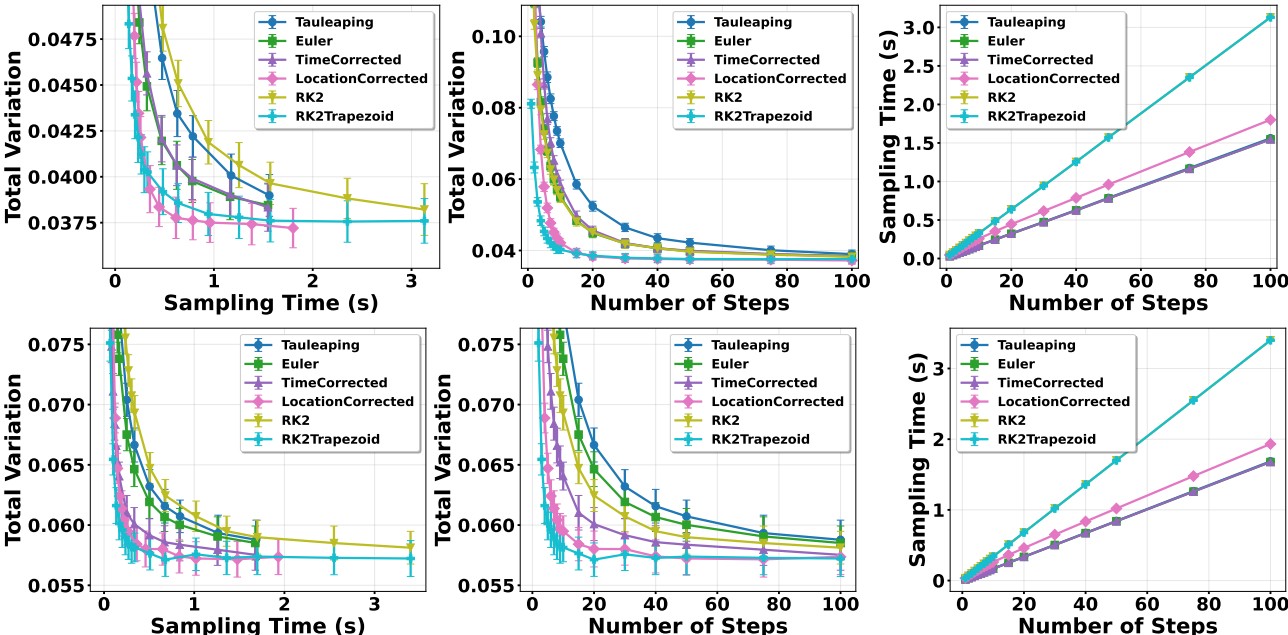

*Figure 2.* Performance comparison of different samplers with 9-dimensional data distribution (vocabulary size is 8). (Top) Masked source distribution; (Bottom) uniform source distribution. Under a fixed sampling time, the location-corrected sampler outperforms other samplers for masked source distribution, and remains competitive with RK2-Trapezoid sampler for uniform source distribution. For a fixed number of steps, the time-corrected sampler has the same sampling time as the tau-leaping and Euler samplers, and the location-corrected sampler has negligible additional cost relative to the tau-leaping and Euler samplers.

this experiments, we consider both masked and uniform source distributions, and use the linear schedule $\kappa_t = t$. We trained our logits models using objective in equation 16. We compared the corrected samplers (Algorithms 4 and 5) with tau-leaping, Euler samplers (Algorithms 2 and 3) and (deterministic midpoint) high-order samplers (Ren et al., 2025b). For a fair comparison, we record the sampling time of each sampler in each setting. The results are provided in Figure 2, demonstrating that the proposed corrected samplers have a faster convergence rate than tau-leaping and Euler samplers. Specifically, the corrected samplers achieves lower total variation when the sampling time is fixed. Furthermore, the corrected samplers have negligible additional cost relative to the Euler sampler when the number of steps is greater than dimension $\mathcal{D}$. Compared with (deterministic midpoint) high-order samplers, the corrected samplers consistently outperform the RK2 sampler. Moreover, under a fixed sampling time, the location-corrected sampler achieves lower total variation than the RK2-Trapezoid sampler for the masked source distribution, and remains competitive with the RK2-Trapezoid sampler for the uniform source distribution. The observed advantage of the location-corrected sampler under the masked source distribution is consistent with our theoretical result that it has lower iteration complexity if the posterior is time-independent.

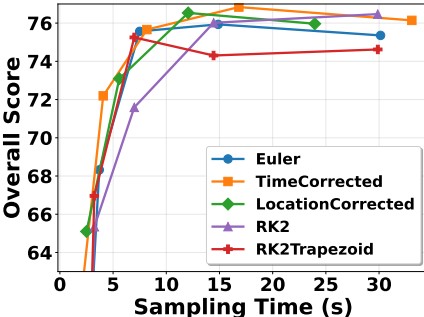

*Figure 3.* GenEval scores of different samplers in text-to-image generation tasks.

### 6.2. Text-to-Image Generation

On the image generation task, we evaluate corrected samplers in a high-dimensional setting ($\mathcal{D} = 576$). In such high-dimensional scenario, using one function evaluation to update one token with location-corrected sampler is computationally inefficient, since the number of steps ($K \in \{4, 8, 16, 32, 64\}$ in our experiments) is typically much smaller than $\mathcal{D}$. To address this issue, we extend our corrected samplers to few-step regime and general conditional rates; see Sections D.3 and D.4. In this experiment, we use FUDOKI (Wang et al., 2025) as our logits models and consider a metric-induced probability path introduced in Shaul et al. (2025). We compare our proposed samplers with the commonly used Euler solver (Algorithm 6; see also

Algorithm 1 in Shaul et al., 2025) and (deterministic mid-point) high-order samplers (Ren et al., 2025b). The GenEval overall scores of different samplers are reported in Figure 3. We observe that the corrected samplers consistently outperforms the Euler solver and RK2-Trapezoid sampler, and converges faster than RK2 sampler, demonstrating that the corrected samplers achieves higher accuracy and sampling efficiency in high-dimensional tasks. In particular, the time-corrected sampler performs well on this task, achieving a higher GenEval score than all other samplers under the same time budget at a sampling time comparable to that of the Euler sampler with $K = 8$.

## 7. Related Work

**Discrete diffusion models and discrete flow models.** Discrete diffusion models (Austin et al., 2021; Hoogeboom et al., 2022; Campbell et al., 2022; Sun et al., 2023) have become increasingly popular in recent years. Meng et al. (2022) and Lou et al. (2024) provide a unified framework for modeling the time-reversal of a forward continuous-time Markov chain (CTMC) by learning a concrete score, parallel to the score matching in continuous diffusion models (Song et al., 2021). Instead of learning concrete score in diffusion-based models, discrete flow-based models (Gat et al., 2024; Shaul et al., 2025; Campbell et al., 2024) offers a flexible framework to train a transition rate that can generate a probability flow from a simple distribution to the data distribution, which incurs zero initialization error compared to concrete score matching. Our work focuses mainly on the discretization error in the sampling stage of discrete flow models.

**Theoretical analysis of discrete diffusion models and discrete flow models.** There are some works on the theoretical results for uniformization algorithm (Chen & Ying, 2024; Huang et al., 2025; Ren et al., 2025a; Wan et al., 2025b), which enables us to simulate CTMC without discretization error. Zhang et al. (2025) establishes error bounds for uniformization algorithm using a piece-wise time-homogeneous transition rate. However, the uniformization algorithm is not efficient since the number of functions evaluation might be large in some time interval. Tau-leaping algorithm (Gillespie, 2001; Campbell et al., 2022) enables all coordinates to update in parallel by freezing both time and location, which leads to a Lévy process (Ren et al., 2025a) and requires only one function call in each time interval. Some theoretical results of tau-leaping are established in the discrete diffusion framework (Ren et al., 2025a; Liang et al., 2025a;b). Ren et al. (2025b) investigates high-order tau-leaping methods (Hu et al., 2011) in discrete diffusion models. For discrete flow models, Su et al. (2025) and Wan et al. (2025b) derive the estimation errors for different empirical risk minimizers and network classes. However, existing works often impose conditions on the

support of data distribution and boundedness of concrete score and transition rate, or focus only on a specific source distribution. In our work, we establish a non-asymptotic error bound of tau-leaping and Euler sampler for discrete flow models in a unified framework by constructing an auxiliary process in our analysis. Additionally, we propose two corrected samplers to reduce the discretization error from the Euler sampler by analyzing its one-step lower bound.

## 8. Conclusion and Future Work

In this work, we study the corrected samplers by considering a two-stage transition rate for jump processes. Compared to tau-leaping and the Euler sampler, the time-corrected sampler reduces the error caused by freezing the time variable of the time schedule in the transition rate, while the location-corrected sampler reduces the iteration complexity of the time-corrected sample by reevaluating network function after the first jump in each timestep. The proposed samplers are more accurate than tau-leaping and the Euler solver and incur almost no additional computational cost. We have rigorously established the non-asymptotic error bounds for the proposed samplers under the assumption of finite estimation error.

As mentioned in Section 5, when the posterior is time-dependent, the theoretical error bound may be dominated by the time-Lipschitz constant of the posterior. We believe that it would be a promising direction to model the *average transition rate* $\frac{1}{\tau_k} \int_{t_{k-1}}^{t_k} Q_s(x, z) \mathrm{d}s$ (similar to the MeanFlow framework of Geng et al., 2025) instead of modeling the *instantaneous* posterior $p_{1|t}$, to control the error by approximating the distribution of the exit time (Proposition 2). On the other hand, the error bounds derived in existing works focus only on the scenario where the number of steps $K$ is sufficiently large; it would be interesting to investigate theoretical results of the proposed samplers in a few-step regime (see, Section D.3).

## Acknowledgements

Hongyuan Zha's research is supported in part by Shenzhen Stability Science Program 2023, and National Natural Science Foundation of China (72495131). Fang Fang gratefully acknowledges research support from National Natural Science Foundation of China (72331005). Guang Cheng gratefully acknowledges financial support through a gift from Cisco.

## Impact Statement

This paper presents work whose goal is to advance the field of machine learning. There are many potential societal consequences of our work, none of which we feel must be specifically highlighted here.

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

# Appendix

The appendix is organized as follows. We introduce some notations in Section A. Section B presents several sampling algorithms discussed in this paper. Section C provides some theoretical results for (two-stage) jump processes. Some discussions and extension of the corrected samplers are presented in Section D. We present auxiliary lemmas and technical proofs in Sections E to G. The implementation details for experiments and additional results are reported in Section H.

## A. Notation

Let $[N] = \{1, 2, \ldots, N\}$ for a positive integer $N$. We use $\dot{\kappa}_t$ to denote the time derivative of a function $\kappa_t$ of $t$. For a $\mathcal{D}$-dimensional vector $z$, denote $z^d$ and $z^{\backslash d}$ as the $d$-th element of the vector $z$ and the $(\mathcal{D}-1)$-dimensional vector $(z^1, \ldots, z^{d-1}, z^{d+1}, \ldots, z^{\mathcal{D}})^\top$. We use $\mathbb{1}(\cdot)$ to denote an indicator function. We denote the Hamming distance of two vectors $z, x$ by $d^H(z, x)$. For two quantities $x, z$, denote $\delta_x(z)$ as the Kronecker delta that satisfies $\delta_x(z) = 1$ if $x = z$ and $\delta_x(z) = 0$ if $x \neq z$. We say $a = \mathcal{O}(b)$ if $a \leq cb$ for a positive constant $c$. We use $a \vee b$ to denote $\max\{a, b\}$.

## B. Algorithms

In this section, we present several sampling algorithms in our analysis. Algorithm 1 describes the uniformization algorithm, which enables us to simulate CTMC in an exact way (Chen & Ying, 2024). The tau-leaping algorithm (Gillespie, 2001) for categorical data is presented in Algorithm 2. We present the Euler sampler, as well as our proposed time-corrected and location-corrected samplers in Algorithms 3 to 5, respectively.

---

**Algorithm 1** Uniformization

---

**Require:** A transition rate $Q_t$, an early stopping parameter $\delta > 0$, time partition $0 = t_0 < t_1 < \cdots < t_K = 1 - \delta$, parameters $\lambda_1, \lambda_2, \ldots, \lambda_K$ satisfying $\sup_{t \in [t_{k-1}, t_k]} \sum_{z \neq x} Q_t(x, z) \leq \lambda_k$ for any $k \in [K], x \in \mathcal{S}^{\mathcal{D}}$.
1: Draw $Y_0 \sim p_0$.
2: **for** $k = 1$ to $K$ **do**
3:     Draw $M \sim \text{Poisson}(\tau_k \lambda_k)$.
4:     Sample $M$ points i.i.d. from $\mathcal{U}([t_{k-1}, t_k])$ and sort them as $s_1 < s_2 < \cdots < s_M$.
5:     Set $Z_0 = Y_{k-1}$.
6:     **for** $j = 1$ to $M$ **do**
7:         Set $Z_j = \begin{cases} z, & \text{with probability } Q_{s_j}(Z_{j-1}, z)/\lambda_k \\ Z_{j-1}, & \text{with probability } 1 - (\sum_{z \neq Z_{j-1}} Q_{s_j}(Z_{j-1}, z)/\lambda_k) \end{cases}$, where $z \neq Z_{j-1}$.
8:     **end for**
9:     Set $Y_k = Z_M$.
10: **end for**
11: **return** $Y_K \sim p_{1-\delta}$

---

---

**Algorithm 2** Tau-leaping

---

**Require:** A transition rate $Q_t$, an early stopping parameter $\delta > 0$, time partition $0 = t_0 < t_1 < \cdots < t_K = 1 - \delta$.
1: Draw $Y_0 \sim p_0$.
2: **for** $k = 1$ to $K$ **do**
3:     **for** $d = 1$ to $\mathcal{D}$ **do**                                                          ▷ in parallel
4:         **for** $z^d \in \mathcal{S} \backslash \{Y_{k-1}^d\}$ **do**
5:             Draw $M_{d, z^d} \sim \text{Poisson}(\tau_k Q_{t_{k-1}}^d(Y_{k-1}, z^d))$.
6:         **end for**
7:     **end for**
8:     Set $Z = Y_{k-1} + \sum_{d=1}^{\mathcal{D}} \sum_{z^d \in \mathcal{S} \backslash \{Y_{k-1}^d\}} (z^d - Y_{k-1}) M_{d, z^d}$.
9:     Set $Y_k = \mathcal{M}_{Y_{k-1}}(Z)$, where $\mathcal{M}_x(z) = \{z^d \delta_{\mathcal{S}}(z^d) + x^d(1 - \delta_{\mathcal{S}}(z^d))\}_{d \in [\mathcal{D}]}$ is a mapping from $\mathbb{Z}^{\mathcal{D}}$ to $\mathcal{S}^{\mathcal{D}}$.
10: **end for**
11: **return** $Y_K \sim p_{1-\delta}$

---

---

**Algorithm 3** Euler sampler

---

**Require:** A transition rate $Q_t$, an early stopping parameter $\delta > 0$, time partition $0 = t_0 < t_1 < \cdots < t_K = 1 - \delta$.

1: Draw $Y_0 \sim p_0$.
2: **for** $k = 1$ to $K$ **do**
3:     Set $Z = Y_{k-1}$.
4:     **for** $d = 1$ to $\mathcal{D}$ **do**                                                        $\triangleright$ in parallel
5:         Set $\lambda_k^d = \sum_{z^d \neq Y_{k-1}^d} Q_{t_{k-1}}^d(Y_{k-1}, z^d)$
6:         Set $Z^d = \begin{cases} z^d, & \text{with probability } \left\{ 1 - \exp\left( -(t_k - t_{k-1})\lambda_k^d \right) \right\} Q_{t_{k-1}}^d(Y_{k-1}, z^d) / \lambda_k^d \\ Z^d, & \text{with probability } \exp\left( -(t_k - t_{k-1})\lambda_k^d \right) \end{cases}$, where $z^d \neq Z^d$.
7:     **end for**
8:     Set $Y_k = Z$.
9: **end for**
10: **return** $Y_K \sim p_{1-\delta}$

---

---

**Algorithm 4** Time-corrected sampler

---

**Require:** A posterior $p_{1|t}$, time schedule $\kappa_t$, an early stopping parameter $\delta > 0$, time partition $0 = t_0 < t_1 < \cdots < t_K = 1 - \delta$.

1: Draw $Y_0 \sim p_0$.
2: **for** $k = 1$ to $K$ **do**
3:     Set $Z = Y_{k-1}$.
4:     **for** $d = 1$ to $\mathcal{D}$ **do**                                                        $\triangleright$ in parallel
5:         Set $\lambda_k^d = \sum_{z^d \neq Y_{k-1}^d} p_{1|t_{k-1}}^d(z^d | Y_{k-1})$
6:         Set $Z^d = \begin{cases} z^d, & \text{with probability } \left\{ 1 - \left( \frac{1 - \kappa_{t_k}}{1 - \kappa_{t_{k-1}}} \right)^{\lambda_k^d} \right\} p_{1|t_{k-1}}^d(z^d | Y_{k-1}) / \lambda_k^d \\ Z^d, & \text{with probability } \left( \frac{1 - \kappa_{t_k}}{1 - \kappa_{t_{k-1}}} \right)^{\lambda_k^d} \end{cases}$, where $z^d \neq Z^d$.
7:     **end for**
8:     Set $Y_k = Z$.
9: **end for**
10: **return** $Y_K \sim p_{1-\delta}$

---

---

**Algorithm 5** Location-corrected sampler

---

**Require:** A posterior $p_{1|t}$, time schedule $\kappa_t$, an early stopping parameter $\delta > 0$, time partition $0 = t_0 < t_1 < \cdots < t_K = 1 - \delta$.

1: Draw $Y_0 \sim p_0$.
2: **for** $k = 1$ to $K$ **do**
3:  Set $\lambda_k = \sum_{d=1}^{\mathcal{D}} \sum_{z^d \neq Y_{k-1}^d} p_{1|t_{k-1}}^d(z^d|Y_{k-1})$
4:  Set $e_k \sim \text{Exp}(\lambda_k)$
5:  Set $T_k = \kappa^{-1}\Big((1 - \kappa_{t_{k-1}})(1 - \exp(-e_k)) + \kappa_{t_{k-1}}\Big)$
6:  **if** $T_k \geq t_k$ **then**
7:   Set $Y_k = Y_{k-1}$
8:  **else**
9:   Set $Y_{k-1} = z$ with probability $\sum_{d=1}^{\mathcal{D}} \delta_{Y_{k-1}^{\backslash d}}(z^{\backslash d}) p_{1|t_{k-1}}^d(z^d|Y_{k-1})/\lambda_k$,
10:   Set $Z = Y_{k-1}$.
11:   **for** $d = 1$ to $\mathcal{D}$ **do**                                                         ▷ in parallel
12:    Set $\lambda_k^d = \sum_{z^d \neq Y_{k-1}^d} p_{1|T_k}^d(z^d|Y_{k-1})$
13:    Set $Z^d = \begin{cases} z^d, & \text{with probability } \left\{1 - \left(\frac{1-\kappa_{t_k}}{1-\kappa_{T_k}}\right)^{\lambda_k^d}\right\} p_{1|T_k}^d(z^d|Y_{k-1})/\lambda_k^d \\ Z^d, & \text{with probability } \left(\frac{1-\kappa_{t_k}}{1-\kappa_{T_k}}\right)^{\lambda_k^d} \end{cases}$, where $z^d \neq Z^d$.
14:   **end for**
15:   Set $Y_k = Z$.
16:  **end if**
17: **end for**
18: **return** $Y_K \sim p_{1-\delta}$

---

## C. Results for Jump Processes

We first define the random measure associated with jump processes as follows.

**Definition 2** (Random Measure). *Define the random measure associated with jump process $X(t)$ on the finite state space $\mathcal{S}^{\mathcal{D}}$ as*

$$N((t_1, t_2], A) = \#\{t_1 < s \leq t_2; \Delta X(s) \neq 0, X(s) \in A\},$$

*where $\Delta X(t) = X(t) - X(t-)$ and $A \subseteq \mathcal{S}^{\mathcal{D}}$. We denote $N(t, A) \triangleq N((0, t], A)$.*

In the rest of this section, we introduce some basic tools required for our analysis.

### C.1. Uniformization

Note that this two-stage transition rate $Q_t(x, z)$ is a predictable process for fixed $x, z \in \mathcal{S}^{\mathcal{D}}$; that is, $Q_t(x, z) \in \mathcal{F}_{t-}$. Similar to CTMC, we can use the following uniformization (Wan et al., 2025b; Chen & Ying, 2024) technique to simulate the jump process $X(t)$ in an exact way.

**Proposition 3** (Uniformization). *Suppose that $T_1, T_2, \ldots$ are the arrival times of a Poisson process $N(t)$ with rate $M$. Let $X(t)$ be a jump process with initial distribution $p_0$ and natural filtration $\mathcal{F}_t$ such that at $t = T_1, T_2, \ldots$, the process $X(t)$ jumps to position $z \neq X(t-)$ with probability $Q_t(X(t-), z)/M$, where $Q_t(x, z)$ is defined in Equation 5 and satisfies that $-Q_t(X(t-), X(t-)) \leq M$ a.s..*

*Then the compensated random measure associated with $X(t)$ is*

$$\tilde{N}(t, A) = N(t, A) - \int_0^t \sum_{z \in A \backslash \{X(s-)\}} Q_s(X(s-), z) ds.$$

By uniformization, the process $X(t)$ has only finite jump times on $[0, t]$ almost surely since $N(t, \mathcal{S}^{\mathcal{D}}) \leq N(t) < \infty$ a.s..

Then $X(t)$ can be written as a stochastic integral with the integrator $N(\mathrm{d}t, \mathrm{d}z)$:

$$X(t) = X(0) + \int_0^t \int_{\mathcal{S}^{\mathcal{D}}} (z - X(s-)) N(\mathrm{d}s, \mathrm{d}z), \tag{15}$$

## C.2. Change of Measure

Next, we will present a Girsanov-type theorem for jump processes, which allows us to derive the KL divergence of path measures of two jump processes with different transition structure.

Suppose that $X(t)$ is a jump process with compensated random measure

$$\tilde{N}^{\mathbb{P}}(t, A) = N(t, A) - \int_0^t \sum_{z \in A \setminus \{X(s-)\}} Q_s^{\mathbb{P}}(X(s-), z) \mathrm{d}s$$

under the probability space $(\Omega, \mathcal{F}, \mathbb{P})$, where $Q_t^{\mathbb{P}}$ is a two-stage transition rate defined in Equation 5 satisfying $-Q_t^{\mathbb{P}}(X(t-), X(t-)) \leq M$ $\mathbb{P}$-a.s.. We aim to find a probability measure $\mathbb{Q}$ such that $X(t)$ is a jump process with compensated random measure

$$\tilde{N}^{\mathbb{Q}}(t, A) = N(t, A) - \int_0^t \sum_{z \in A \setminus \{X(s-)\}} Q_s^{\mathbb{Q}}(X(s-), z) \mathrm{d}s$$

under the probability space $(\Omega, \mathcal{F}, \mathbb{Q})$, where $\mathbb{Q}$ is a two-stage transition rate. Similar to section 4 in Wan et al. (2025b), we define $Q \ll \mathbb{P}$ such that $\frac{\mathbb{Q}_{0:t}}{\mathbb{P}_{0:t}} = \exp(W(t))$, where the path measure $\mathbb{P}_{0:t}$ is the restriction of $\mathbb{P}$ on $(\Omega, \mathcal{F}_t)$ and

$$W(t) = \int_0^t \sum_{z \neq X(s-)} \log \frac{Q_s^{\mathbb{Q}}(X(s-), z)}{Q_s^{\mathbb{P}}(X(s-), z)} N(\mathrm{d}s, z) + \int_0^t \sum_{z \neq X(s-)} [Q_s^{\mathbb{P}}(X(s-), z) - Q_s^{\mathbb{Q}}(X(s-), z)] \mathrm{d}s.$$

**Theorem 5** (Change of Measure). *Assume that $Q_s^{\mathbb{P}}(X(s-), z) = 0$ implies $Q_s^{\mathbb{Q}}(X(s-), z) = 0$ for any $s \in [0, \tau]$ and $z \in \mathcal{S}^{\mathcal{D}}$ $\mathbb{P}$-a.s.. Then $\exp(W(t))$ is a $\mathbb{P}$-martingale on $[0, \tau]$, and $\tilde{N}^Q(t, A)$ is the compensated random measure associated with $X(t)$ under probability space $(\Omega, \mathcal{F}, \mathbb{Q})$. Moreover, we have*

$$D_{KL}(\mathbb{Q}_{1:t} || \mathbb{P}_{1:t}) = \mathbb{E}_{\mathbb{Q}} \Big[ \int_0^t \sum_{z \neq X(s)} D_F \Big( Q_s^{\mathbb{Q}}(X(s), z) || Q_s^{\mathbb{P}}(X(s), z) \Big) ds \Big],$$

*where $D_F$ is the Bregman divergence induced by $F(x) = x \log x$.*

By utilizing Theorem 5 and Jensen's inequality, we derive the bound for the KL divergence of marginals of two jump processes:

$$D_{\mathrm{KL}}(p_t^{\mathbb{Q}} || p_t^{\mathbb{P}}) \leq \mathbb{E}_{\mathbb{Q}} \Big[ \int_0^t \sum_{z \neq X(s)} D_F \Big( Q_s^{\mathbb{Q}}(X(s), z) || Q_s^{\mathbb{P}}(X(s), z) \Big) \mathrm{d}s \Big].$$

# D. Discussion

## D.1. Training Objective in Terms of Posterior

To derive the training objective in terms of the posterior $p_{1|t}$, one can plug equation 1 and equation 2 and equation 3 into equation 4:

$$\mathbb{E} \Big[ \frac{\dot{\kappa}_t}{1 - \kappa_t} \sum_{d=1}^{\mathcal{D}} \Big\{ - \Big( 1 - \delta_{X^d(1)}(X^d(\mathbf{t})) \Big) \log p_{1|t}(X^d(1)|X(\mathbf{t})) + \delta_{X^d(1)}(X^d(\mathbf{t})) - p_{1|t}(X^d(\mathbf{t})|X(\mathbf{t})) \Big\} \Big], \tag{16}$$

which is equivalent to Equation 37 in (Shaul et al., 2025). If we use the masked source distribution with the mixture path, then we have $\delta_{X^d(1)}(X^d(\mathbf{t})) = 1 - \delta_{\mathrm{m}}(X^d(\mathbf{t}))$, and the underlying posterior satisfies $p_{1|t}^d(x^d|x) = (1 - \delta_{\mathrm{m}}(x^d))$, which implies that $\delta_{X^d(1)}(X^d(\mathbf{t})) = p_{1|t}^d(X^d(\mathbf{t})|X(\mathbf{t}))$. In this case, equation 16 recovers the training objective for masked diffusion models (Shi et al., 2024; Sahoo et al., 2024; Ou et al., 2025; Nie et al., 2025).

**Remark 6.** *In the training stage, we also use the early stopping strategy, since it is hard to control the estimation error if $\delta = 0$ (the expectation $\mathbb{E}[\frac{\dot{\kappa}_{\mathbf{t}}}{1 - \kappa_{\mathbf{t}}}]$ is infinite if $\mathbf{t} \sim \mathcal{U}([0, 1])$).*

## D.2. Error Bound with Lower-Boundedness Condition

In this subsection, we establish the total variation error bound for our proposed location-corrected sampler under a lower-boundedness condition for the posterior.

**Assumption 2** (Boundedness). *The posterior satisfies that $p_{1|t}^d(z^d|x) > \underline{M}$ for any $t \in [0, 1 - \delta], x \in \mathcal{S}^{\mathcal{D}}, d \in \mathcal{D}, z^d \neq x^d$.*

This assumption provides strong convexity for the Bregman divergence. Since we only impose condition on the posterior instead of the oracle rate, this assumption is weaker than Assumption 1 in Wan et al. (2025b). Specifically, this assumption could hold for time schedulers with $\lim_{t \to 0^-} \dot{\kappa}_t = 0$ (e.g. consine scheduler $\cos^2(\frac{\pi}{2}(1 - t))$) compared to a condition on the oracle rate (Assumption 1 in Wan et al. (2025b)).

**Theorem 6.** *Assume that $\max_{k \in [K]} \{\tau_k \mathcal{D} M_k\} \leq \log 2$, where $M_k = \sup_{t \in [t_{k-1}, t_k]} |\dot{\kappa}_t/(1 - \kappa_t)|$. The location-corrected sampler has the following bound for the total variation between the estimated and the oracle marginal distribution at time $t = 1 - \delta$:*

$$\mathbb{E}[TV(p_{1-\delta}, \hat{p}_{1-\delta}^{LC})] \leq \sqrt{2\mathcal{D}^3|\mathcal{S}| \sum_{k=1}^{K} \tau_k^3 M_k^3 \underline{M}_k^{-1} \log(\mathcal{D} M_k \tau_k)^{-2}} + \varepsilon_{Est}^{LC}$$

*where $\underline{M}_k = \inf_{t \in [t_{k-1}, t_k], x \in \mathcal{S}^{\mathcal{D}}, d \in \mathcal{D}, z^d \in x^d} \{p_{1|t}^d(z^d|x)\}$, and $\varepsilon_{Est}^{LC}$ is defined in Assumption 1. In particular, under Assumption 2, it holds that*

$$\mathbb{E}[TV(p_{1-\delta}, \hat{p}_{1-\delta}^{LC})] \leq \sqrt{2\mathcal{D}^3|\mathcal{S}|\underline{M}^{-1} \sum_{k=1}^{K} \tau_k^3 M_k^3 \log(\mathcal{D} M_k \tau_k)^{-2}} + \varepsilon_{Est}^{LC}.$$

This error bound does not depend on the time-Lipschitz constant $L_k^p$ used in Theorem 4. Compared to the error bound for the time-independent posterior given in Theorem 4, the sacrifice incurred by the time-Lipschitz constant of the posterior is $\underline{M}_k$ in each time interval.

Consider a uniform source distribution with mixture path; that is $p_0^d(x^d) = 1/|\mathcal{S}|$ for all $d \in \mathcal{D}$ and $x^d \in \mathcal{S}$. Similar to Appendix B.2 in Wan et al. (2025b), we can further derive an explicit time-independent lower bound for the posterior

$$
\begin{aligned}
p_{1|t}^d(z^d|x) &= \frac{p_{t|1}^d(x^d|z^d) \sum_{z \backslash} \prod_{i \neq d} p_{t|1}^i(x^i|z^i) p_1(z)}{\sum_y p_{t|1}^d(x^d|y^d) \prod_{i \neq d} p_{t|1}^i(x^i|y^i) p_1(y)} \\
&= \frac{\frac{1-\kappa_t}{|\mathcal{S}|} p_1^d(z^d) \sum_{z \backslash d} \prod_{i \neq d} p_{t|1}^i(x^i|z^i) p_1(z^{\backslash d}|z^d)}{\frac{1-\kappa_t}{|\mathcal{S}|} \sum_{y \backslash d} \prod_{i \neq d} p_{t|1}^i(x^i|y^i) p_1(y^{\backslash d}) + \kappa_t p_1^d(x^d) \sum_{y \backslash d} \prod_{i \neq d} p_{t|1}^i(x^i|y^i) p_1(y^{\backslash d}|x^d)} \\
&\geq \frac{(1 - \kappa_t)\alpha\beta^{-1}}{(1 - \kappa_t) + \kappa_t \alpha^{-1}|\mathcal{S}|} \\
&\geq \frac{\delta\alpha\beta^{-1}}{1 + \alpha^{-1}|\mathcal{S}|},
\end{aligned}
$$

where $t \in [0, 1 - \delta]$, $\alpha = \inf_{x_1 \in \mathcal{S}^{\mathcal{D}}, d \in [\mathcal{D}]} \min \{\frac{p_1(x_1^{\backslash d}|x_1^d)}{p_1(x_1^{\backslash d})}, \frac{p_1(x_1^{\backslash d})}{p_1(x_1^{\backslash d}|x_1^d)}\}$ and $\beta = \sup_{d, z^d, x^d}(p_1^d(z^d)/p_1^d(x^d))$. Some discussions and examples of $\alpha$ and $\beta$ can be found in the Appendix B.2 in Wan et al. (2025b).

## D.3. Discussion on Few-Step Regime

When the stepsize $\tau_k$ and the dimension $\mathcal{D}$ are relatively large, the condition $\max_{k \in [K]} \{\tau_k \mathcal{D} M_k\} \leq 1$ in Theorems 1, 3 and 4 does not hold. Therefore, by Proposition 2, the probability of a jump occurring in each time interval is large. In this case, using one function evaluation to update one token of a $\mathcal{D}$-dimensional sequence is inefficient for the location-corrected sampler (Algorithm 5). To handle this case, we can consider the $j$-th arrival time in the two-stage transition rate (equation 5); that is, for $j \in \mathbb{Z}_+$ and $z \neq x$,

$$Q_t^{LC,(j)}(x, z) = \mathbb{1}(T_k^{(j)} \geq t) \underbrace{Q_{1,t}^{TC}(x, z)}_{\text{first-stage rate}} + \mathbb{1}(T_k^{(j)} < t) \underbrace{Q_{2,t}(x, z; X(T_k^{(j)}), T_k^{(j)})}_{\text{second-stage rate}}, \ t \in [t_{k-1}, t_k] \tag{17}$$

where $T_k^{(j)} = \inf \{t > T_k^{(j-1)} : \Delta X(t) \neq 0\}$ $(T_k^{(0)} = t_{k-1})$,

$$Q_{1,t}^{\text{LC}}(x, z) = \frac{\dot{\kappa}_t}{1 - \kappa_t} \sum_{d=1}^{\mathcal{D}} \delta_{x^{\backslash d}}(z^{\backslash d}) p_{1|t_{k-1}}^d(z^d|x);$$

$$Q_{2,t}^{\text{LC}}(x, z|X(T_k^{(j)}), T_k^{(j)}) = \frac{\dot{\kappa}_t}{1 - \kappa_t} \sum_{d=1}^{\mathcal{D}} \delta_{x^{\backslash d}}(z^{\backslash d}) \delta_{X^d(T_k^{(j)})}(x^d) p_{1|T_k^{(j)}}^d(z^d|X(T_k^{(j)})).$$

In particular, when $j = 1$, the above transition rate is equal to equation 11. The location-corrected sampler associated with the transition rate equation 17 enables us to update $j$ tokens in parallel when the number of steps $K$ is small.

**Remark 7.** *There are various choice of the stopping time in the two-stage transition rate. For example, instead of considering the $j$-th arrival time in the interval $[t_{k-1}, t_k]$, we can also use the first arrival time in the interval $[t_{k-1} + \theta \tau_k, t_k]$, where $\theta \in (0, 1)$.*

From equation 17, the event that location-correction appears in the $k$-th interval depends on the choice of $j$. Suppose that the posterior is time-independent and $T_k^{(j)} < t_k$. Then, when $t \in [t_{k-1}, T_k^{(1)}) \cup [T_k^{(j)}, T_k^{(j+1)} \wedge t_k)$, we have $Q_t^{\text{LC}}(X(t), z) = Q_t(X(t), z)$ for $z \neq X(t)$, where $Q_t$ is the transition rate associated with the idealized process. Consequently, we can reduce the disretization error under the event $\{T_k^{(j)} < t_k\}$; see Figure 4.

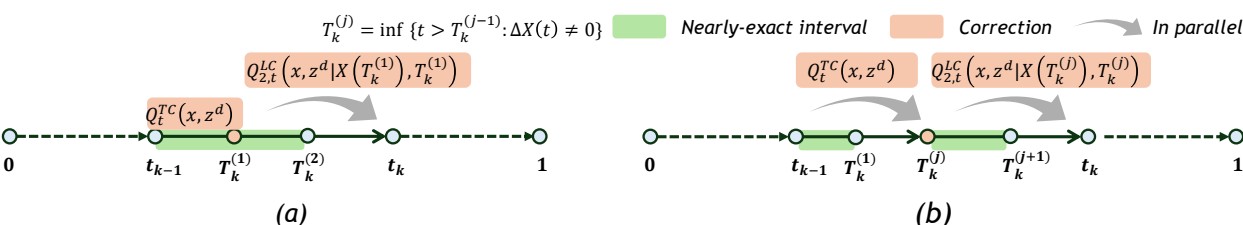

*Figure 4.* Location-corrected sampler for few-step regime. (a) the transition structure in equation 11; (b) the transition structure in equation 17.

### D.4. Discussion on General Conditional Rates

In this subsection, we consider a general conditional transition rate $Q_t^d(x^d, z^d|x_1^d)$, including the kinetic optimal rate used in Shaul et al. (2025); Wang et al. (2025); Wan et al. (2025a) and a family of conditional rates constructed by a detailed balance condition (see, Proposition 3 in Campbell et al., 2024).

**Euler sampler.** Note that the unconditional rate at time $t$ can be written as $Q_t^d(x, z^d) = \mathbb{E}_{\mathbf{x}_1^d \sim p_{1|t}^d(\cdot|x)}[Q_t^d(x, z^d|\mathbf{x}_1^d)]$ for each $d \in [\mathcal{D}]$. To estimate this transition rate, a practical approach is to sample $\mathbf{x}_1^d \sim p_{1|t}^d(\cdot|x)$ and to use approximation $Q_t^d(x, z^d) \approx Q_t^d(x^d, z^d|\mathbf{x}_1^d)$, which results in the always-valid Euler solver introduced in Shaul et al. (2025) with the following rate:

$$Q_t^{\text{Euler}}(x, z) = \sum_{d=1}^{\mathcal{D}} \delta_{x_{k-1}^{\backslash d}}(z^{\backslash d}) \delta_{x_{k-1}^d}(x^d) Q_{t_{k-1}}^d(x_{k-1}^d, z^d|\mathbf{x}_1^d),$$

where $t \in [t_{k-1}, t_k]$ and $\mathbf{x}_1^d \sim p_{1|t_{k-1}}^d(\cdot|x_{k-1})$. The Euler sampler for a general conditional rate is described in Algorithm 6.

**Time-corrected sampler.** Parallel to the time-corrected sampler proposed in Section 4, we consider the following transition rate:

$$Q_t^{\text{TC}}(x, z) = \sum_{d=1}^{\mathcal{D}} \delta_{x_{k-1}^{\backslash d}}(z^{\backslash d}) \delta_{x_{k-1}^d}(x^d) Q_t^d(x_{k-1}^d, z^d|\mathbf{x}_1^d),$$

where $t \in [t_{k-1}, t_k]$ and $\mathbf{x}_1^d \sim p_{1|t_{k-1}}^d(\cdot|x_{k-1})$. Compared to the Euler sampler, the transition rate of the time-corrected sampler does not fix the time variable in the conditional transition rate. To derive the associated algorithm, by Proposition 2,

---

**Algorithm 6** Euler sampler (general conditional rate; Algorithm 1 in Shaul et al., 2025)

---

**Require:** A conditional rate $Q_t^d(x^d, z^d | x_1^d)$, a posterior $p_{1|t}$, an early stopping parameter $\delta > 0$, time partition $0 = t_0 < t_1 < \cdots < t_K = 1 - \delta$.

1: Draw $Y_0 \sim p_0$.
2: **for** $k = 1$ to $K$ **do**
3:     Set $Z = Y_{k-1}$.
4:     **for** $d = 1$ to $\mathcal{D}$ **do**                                     ▷ in parallel
5:         Sample $\mathbf{x}_1^d \sim p_{1|t_{k-1}}^d(\cdot | Y_{k-1})$.
6:         Set $\lambda_k^d = \sum_{z^d \neq Y_{k-1}^d} Q_{t_{k-1}}^d(Y_{k-1}^d, z^d | \mathbf{x}_1^d)$.
7:         Set $Z^d = \begin{cases} z^d, & \text{with probability } \left\{ 1 - \exp\left( -(t_k - t_{k-1})\lambda_k^d \right) \right\} Q_{t_{k-1}}^d(Y_{k-1}^d, z^d | \mathbf{x}_1^d)/\lambda_k^d \\ Z^d, & \text{with probability } \exp\left( -(t_k - t_{k-1})\lambda_k^d \right) \end{cases}$, where $z^d \neq Z^d$.
8:     **end for**
9:     Set $Y_k = Z$.
10: **end for**
11: **return** $Y_K \sim p_{1-\delta}$

---

we have to calculate the integral $\int_{t_{k-1}}^t Q_s^d(x_{k-1}^d, x_{k-1}^d | \mathbf{x}_1^d) ds$ for each $d \in [\mathcal{D}]$. We can use numerical methods to compute this integral: $\int_{t_{k-1}}^t Q_s^d(x_{k-1}^d, x_{k-1}^d | \mathbf{x}_1^d) ds \approx \frac{\tau_k}{m} \sum_{i=1}^{l-1} Q_{s_j}^d(x_{k-1}^d, x_{k-1}^d | \mathbf{x}_1^d)$, where $m$ is the number of grid points, $s_i = t_{k-1} + \frac{i-1}{m}\tau_k$, and $l = \lfloor \frac{m(t-t_{k-1})}{\tau_k} \rfloor$. The time-corrected sampler for general transition rates can be found in Algorithm 7.

**Remark 8.** *The time-corrected sampler (Algorithm 7) requires only one function call per timestep. Given $\mathbf{x}_1^d$, the computational cost of calculating numerical integral is negligible compared to that of a network evaluation. Additionally, when $m = 1$, the time-corrected sampler reduces to the Euler sampler (Algorithm 6).*

---

**Algorithm 7** Time-corrected sampler (general conditional rate)

---

**Require:** A conditional rate $Q_t^d(x^d, z^d | x_1^d)$, a posterior $p_{1|t}$, the number of grid points for numerical integration $m$, an early stopping parameter $\delta > 0$, time partition $0 = t_0 < t_1 < \cdots < t_K = 1 - \delta$.

1: Draw $Y_0 \sim p_0$.
2: **for** $k = 1$ to $K$ **do**
3:     Set $Z = Y_{k-1}$.
4:     **for** $d = 1$ to $\mathcal{D}$ **do**                                     ▷ in parallel
5:         Sample $\mathbf{x}_1^d \sim p_{1|t_{k-1}}^d(\cdot | Y_{k-1})$.
6:         Calculate $\lambda_{k,i}^d = \sum_{z^d \neq Y_{k-1}^d} Q_{s_j}^d(Y_{k-1}^d, z^d | \mathbf{x}_1^d)$, where $s_i = t_{k-1} + \frac{i-1}{m}(t_k - t_{k-1})$.
7:         Set $(T_k^d, l_k^d) = \begin{cases} (s_l, l), & \text{with probability } \exp\left( -\frac{t_k - t_{k-1}}{m} \sum_{i=1}^{l-1} \lambda_{k,i}^d \right) - \exp\left( -\frac{t_k - t_{k-1}}{m} \sum_{i=1}^{l} \lambda_{k,i}^d \right) \\ (t_k, m+1), & \text{with probability } \exp\left( -(t_k - t_{k-1})\frac{1}{m} \sum_{i=1}^m \lambda_{k,i}^d \right). \end{cases}$
8:         **if** $T_k^d \neq t_k$ **then**
9:             Set $Z^d = z^d$ with probability $\frac{Q_{T_k^d}^d(Y_{k-1}^d, z^d | \mathbf{x}_1^d)}{\lambda_{k,l_k^d}^d}$, where $z^d \neq Z^d$.
10:         **end if**
11:     **end for**
12:     Set $Y_k = Z$.
13: **end for**
14: **return** $Y_K \sim p_{1-\delta}$

---

**Location-corrected sampler.** Similar to the location-corrected sampler introduced in Section 5, we consider the following

(two-stage) transition rate:

$$Q_t^{\text{LC}}(x, z) = \mathbb{1}(T_k^{(j)} \geq t) \underbrace{\sum_{d=1}^{\mathcal{D}} \delta_{x_{k-1}^{\backslash d}}(z^{\backslash d}) \delta_{x_{k-1}^d}(x^d) Q_t^d(x_{k-1}^d, z^d | \mathbf{x}_{1,(1)}^d)}_{\text{first-stage rate}} + \mathbb{1}(T_k^{(j)} < t) \underbrace{\sum_{d=1}^{\mathcal{D}} \delta_{x^{\backslash d}}(z^{\backslash d}) \delta_{X^d(T_k)}(x^d) Q_{T_k}^d(X^d(T_k), z^d | \mathbf{x}_{1,(2)}^d)}_{\text{second-stage rate}},$$

where $t \in [t_{k-1}, t_k]$, $T_k^{(j)}$ is the $j$-th arrival time, $\mathbf{x}_{1,(1)}^d \sim p_{1|t_{k-1}}^d(\cdot | x_{k-1})$, and $\mathbf{x}_{1,(2)}^d \sim p_{1|T_k^{(j)}}^d(\cdot | X(T_k^{(j)}))$. Empirically, we can first calculate the arrival times $\{T_k^d\}_{d \in [\mathcal{D}]}$ for all dimensions in parallel, and then choose the $j$-th smallest element of $\{T_k^d\}_{d \in [\mathcal{D}]}$. The algorithm is presented in Algorithm 8.

**Remark 9.** *To accelerate the algorithm, one can employ location correction after a time threshold $t_\theta \in (0, 1)$, and use the time-corrected sampler before $t_\theta$; see Algorithm 8. This additional parameter would help us balance accuracy and efficiency. We compare the different choice of $t_\theta$ in our image generation task.*

## E. Auxiliary Lemmas

**Lemma 1** (Itô's formula). *Consider the random measure $N(t, A)$ associated with the jump process $X(t)$ in Proposition 3. Define a stochastic integral $W(t) = W(0) + \int_0^t \int_A K(s, z) N(ds, dz)$, where $K$ is a predictable process and $A \subseteq \mathcal{S}^{\mathcal{D}}$. For any function $f : \mathbb{R}^{\mathcal{D}} \to \mathbb{R}, t \geq 0$, we have*

$$f(W(t)) - f(W(0)) = \int_0^t \int_A [f(W(s-) + K(s, z)) - f(W(s-))] N(ds, dz).$$

*Proof.* This proof is similar to Lemma 4.4.5 in (Applebaum, 2009). Let $T_0^A = 0$ and $T_n^A = \inf \{t > T_{n-1}^A : \Delta N(t, A) \neq 0\}$. Note that

$$\begin{aligned}
f(W(t)) - f(W(0)) &= \sum_{n=1}^{\infty} f(W(t \wedge T_n^A)) - f(W(t \wedge T_n^A -)) \\
&= \sum_{n=1}^{\infty} [f(W(t \wedge T_n^A -) + K(t \wedge T_n^A, X(t \wedge T_n^A))) - f(W(t \wedge T_n^A -))] \\
&= \int_0^t \int_A [f(W(s-) + K(s, z)) - f(W(s-))] N(ds, dz),
\end{aligned}$$

which completes the proof. $\square$

**Lemma 2** (Kolmogorov backward equation). *The CTMC $X(t)$ satisfies the following equation:*

$$\dot{p}_{1|t}(z|x) = -\sum_y p_{1|t}(z|y) Q_t(x, y).$$

*Proof.* By Bayes' rule, $p_{1|t}(z|x)$ is continuous in $t$. By C-K equation and the definition of CTMC, we have

$$\begin{aligned}
p_{1|t+h}(z|x) - p_{1|t}(z|x) &= p_{1|t+h}(z|x) - \sum_y p_{1|t+h}(z|y)(\delta_x(y) + Q_t(x, y)h + o(h)) \\
&= -\sum_y p_{1|t+h}(z|y) Q_t(x, y)h + o(h),
\end{aligned}$$

which completes the proof by letting $h \to 0^+$. $\square$

**Lemma 3** (Lipschitz regularity in time). *The time derivative of the oracle transition rate is*

$$\dot{Q}_t(x, z) = \sum_{d=1}^{\mathcal{D}} \delta_{x^{\backslash d}}(z^{\backslash d}) \left\{ \frac{\ddot{\kappa}_t(1 - \kappa_t) + (\dot{\kappa}_t)^2}{(1 - \kappa_t)^2} p_{1|t}^d(z^d|x) - \frac{\dot{\kappa}_t}{1 - \kappa_t} \sum_y p_{1|t}^d(z^d|y) Q_t(x, y) \right\},$$

---

**Algorithm 8** Location-corrected sampler (general conditional rate)

---

**Require:** A conditional rate $Q_t^d(x^d, z^d|x_1^d)$, a posterior $p_{1|t}$, the number of grid points for numerical integration $m$, an early stopping parameter $\delta > 0$, a hyperparameter of location-corrected sampler $j$, a time threshold $t_\theta \in [0, 1-\delta)$, time partition $0 = t_0 < t_1 < \cdots < t_K = 1 - \delta$.

1: Draw $Y_0 \sim p_0$.
2: **for** $k = 1$ to $K$ **do**
3:      **for** $d = 1$ to $\mathcal{D}$ **do**              ▷ in parallel
4:          Sample $\mathbf{x}_1^d \sim p_{1|t_{k-1}}^d(\cdot|Y_{k-1})$.
5:          Calculate $\lambda_{k,i}^d = \sum_{z^d \neq Y_{k-1}^d} Q_{s_j}^d(Y_{k-1}^d, z^d|\mathbf{x}_1^d)$, where $s_i = t_{k-1} + \frac{i-1}{m}(t_k - t_{k-1})$.
6:          Set $(T_k^d, l_k^d) = \begin{cases} (s_l, l), & \text{with probability } \exp\left(-\frac{t_k - t_{k-1}}{m}\sum_{i=1}^{l-1}\lambda_{k,i}^d\right) - \exp\left(-\frac{t_k - t_{k-1}}{m}\sum_{i=1}^{l}\lambda_{k,i}^d\right) \\ (t_k, m+1), & \text{with probability } \exp\left(-(t_k - t_{k-1})\frac{1}{m}\sum_{i=1}^{m}\lambda_{k,i}^d\right). \end{cases}$
7:      **end for**
8:      Set $T_k^{(j)}$ is the $j$-th order statistic of $\{T_k^d\}_{d \in [\mathcal{D}]}$ in ascending order.
9:      **if** $T_k^{(j)} = t_k$ or $t \leq t_\theta$ **then**
10:          **for** $d = 1$ to $\mathcal{D}$ **do**              ▷ in parallel
11:              **if** $T_k^d \neq t_k$ **then**
12:                  Set $Z^d = z^d$ with probability $\frac{Q_{T_k^d}^d(Y_{k-1}^d, z^d|\mathbf{x}_1^d)}{\lambda_{k,l_k^d}^d}$, where $z^d \neq Z^d$.
13:              **end if**
14:          **end for**
15:          Set $Y_k = Z$.
16:      **else**
17:          **for** $d = 1$ to $\mathcal{D}$ **do**              ▷ in parallel
18:              **if** $T_k^d \leq T_k^{(j)}$ **then**
19:                  Set $Z^d = z^d$ with probability $\frac{Q_{T_k^d}^d(Y_{k-1}^d, z^d|\mathbf{x}_1^d)}{\lambda_{k,l_k^d}^d}$, where $z^d \neq Z^d$.
20:              **end if**
21:          **end for**
22:          Set $Y_{k-1} = Z$.
23:          **for** $d = 1$ to $\mathcal{D}$ **do**              ▷ in parallel
24:              Sample $\mathbf{x}_1^d \sim p_{1|T_k^{(j)}}^d(\cdot|Y_{k-1})$.
25:              Calculate $\lambda_{k,i}^d = \sum_{z^d \neq Y_{k-1}^d} Q_{s_j}^d(Y_{k-1}^d, z^d|\mathbf{x}_1^d)$, where $s_i = T_k^{(j)} + \frac{i-1}{m}(t_k - T_k^{(j)})$.
26:              Set $(T_k^d, l_k^d) = \begin{cases} (s_l, l), & \text{with probability } \exp\left(-\frac{t_k - T_k^{(j)}}{m}\sum_{i=1}^{l-1}\lambda_{k,i}^d\right) - \exp\left(-\frac{t_k - T_k^{(j)}}{m}\sum_{i=1}^{l}\lambda_{k,i}^d\right) \\ (t_k, m+1), & \text{with probability } \exp\left(-(t_k - T_k^{(j)})\frac{1}{m}\sum_{i=1}^{m}\lambda_{k,i}^d\right). \end{cases}$
27:              **if** $T_k^d \neq t_k$ **then**
28:                  Set $Z^d = z^d$ with probability $\frac{Q_{T_k^d}^d(Y_{k-1}^d, z^d|\mathbf{x}_1^d)}{\lambda_{k,l_k^d}^d}$, where $z^d \neq Z^d$.
29:              **end if**
30:          **end for**
31:          Set $Y_k = Z$.
32:      **end if**
33: **end for**
34: **return** $Y_K \sim p_{1-\delta}$

---

*which implies that* $|\dot{Q}_t(x,z)| \leq \left|\frac{\ddot{\kappa}_t(1-\kappa_t)+(\dot{\kappa}_t)^2}{(1-\kappa_t)^2}\right| + \mathcal{D}\left(\frac{\dot{\kappa}_t}{1-\kappa_t}\right)^2$ *for any* $z \neq x$. *In particular, if the posterior* $p_{1|t}$ *is not related to* $t$, *then* $|\dot{Q}_t(x,z)| \leq \left|\frac{\ddot{\kappa}_t(1-\kappa_t)+(\dot{\kappa}_t)^2}{(1-\kappa_t)^2}\right|$.

*Proof.* By Kolmogorov backward equation, we have

$$\dot{Q}_t(x,z) = \frac{\mathrm{d}}{\mathrm{d}t}\left(\sum_{d=1}^{\mathcal{D}} \delta_{x^{\setminus d}}(z^{\setminus d})\frac{\dot{\kappa}_t}{1-\kappa_t}p_{1|t}^d(z^d|x)\right)$$

$$= \sum_{d=1}^{\mathcal{D}} \delta_{x^{\setminus d}}(z^{\setminus d})\frac{\ddot{\kappa}_t(1-\kappa_t)+(\dot{\kappa}_t)^2}{(1-\kappa_t)^2}p_{1|t}^d(z^d|x) - \sum_{d=1}^{\mathcal{D}} \delta_{x^{\setminus d}}(z^{\setminus d})\frac{\dot{\kappa}_t}{1-\kappa_t}\sum_y p_{1|t}^d(z^d|y)Q_t(x,y).$$

Consequently, we obtain that

$$|\dot{Q}_t(x,z)| \leq \left|\frac{\ddot{\kappa}_t(1-\kappa_t)+(\dot{\kappa}_t)^2}{(1-\kappa_t)^2}\right| + \mathcal{D}\left(\frac{\dot{\kappa}_t}{1-\kappa_t}\right)^2.$$

Moreover, if the posterior is not related to $t$, then

$$\dot{Q}_t(x,z) = \sum_{d=1}^{\mathcal{D}} \delta_{x^{\setminus d}}(z^{\setminus d})\left\{\frac{\ddot{\kappa}_t(1-\kappa_t)+(\dot{\kappa}_t)^2}{(1-\kappa_t)^2}p_{1|t}^d(z^d|x)\right\},$$

which completes the proof.

$\square$

**Lemma 4** (Property of Bregman divergence). *Suppose that* $D_F(a||b)$ *is the Bregman divergence induced by the function* $F(x) = x\log x$, *where* $a \geq 0$ *and* $b \geq \epsilon$. *Then the Bregman divergence can be bounded by*

$$D_F(a||b) \leq b \vee \left(a\log\frac{a}{\epsilon}\right).$$

*Additionally, we have*

$$D_F(a||b) \leq \frac{(a-b)^2}{2\epsilon} \vee ((b-a)+\epsilon)$$

*Proof.* Note that if $a/b \geq 1$, then we have

$$D_F(a||b) = a\log\frac{a}{b} - a + b \leq a\log\frac{a}{\epsilon},$$

If $a/b < 1$, then we have $D_F(a||b) \leq b$, which proves the first inequality.

On the other hand, by the smoothness of $F(x)$; that is, $F''(x) = 1/x$, if $a > \epsilon$, we have $D_F(a||b) \leq (a-b)^2/(2\epsilon)$; if $a \leq \epsilon \leq b$, then $D_F(a||b) \leq (b-a)+\epsilon$. $\square$

**Lemma 5** (Change of variable). *If* $e_k \sim Exp(\lambda_k)$, *then* $T_k \overset{d}{=} \kappa^{-1}\left((1-\kappa_{t_{k-1}})(1-\exp(-e_k))+\kappa_{t_{k-1}}\right)$, *where* $T_k$ *is of the distribution with survival function in Equation* (12).

*Proof.* Note that $\mathbb{P}(e_k > s) = \exp(-\lambda_k s)$ for $s \geq 0$. Define an increasing function

$$g(x) = \kappa^{-1}\left((1-\kappa_{t_{k-1}})(1-\exp(-x))+\kappa_{t_{k-1}}\right),$$

where $\kappa^{-1}(\cdot)$ is the inverse function of $\kappa_t$. Let $t = g(s)$. Then $s = -\log\frac{1-\kappa_t}{1-\kappa_{t_{k-1}}}$. By change of variable, we have

$$\mathbb{P}(g(e_k) > t) = \mathbb{P}(e_k > s) = \exp(\lambda_k\log\frac{1-\kappa_t}{1-\kappa_{t_{k-1}}}) = \mathbb{P}(T_k > t),$$

which completes the proof. $\square$

# F. Proof of Results for Jump Processes

## F.1. Proof of Proposition 2

*Proof.* Consider a sequence of events $E_{n,i} = \{X(s) = X(\frac{i}{2^n}t) \text{ for any } s \in [\frac{i-1}{2^n}t, \frac{i}{2^n}t]\}$, where $i \in [2^n]$. Then, by uniformization and Equation 18 we have

$$\mathbb{P}(E_{n,i}| \cap_{j=1}^{i-1} E_{n,j}) = \mathbb{P}\Big( N\Big( (\frac{i-1}{2^n}t, \frac{i}{2^n}t], \mathcal{S}^\mathcal{D}\Big) = 0 \Big| \cap_{j=1}^{i-1} E_{n,j}\Big)$$

$$= 1 + \int_{\frac{i-1}{2^n}t}^{\frac{i}{2^n}t} Q_{1,s}(x,x)\mathrm{d}s + R_{n,i},$$

where $|R_{n,i}| \le 3M^2(t/2^n)^2$ for large enough $n$. Consequently, we have

$$\mathbb{P}(T^X > t) = \lim_{n\to\infty} \mathbb{P}(\cap_{i=1}^{2^n} E_{n,i})$$

$$= \lim_{n\to\infty} \prod_{i=1}^{2^n} \Big( \mathbb{P}(E_{n,i}| \cap_{j=1}^{i-1} E_{n,j})\Big)$$

$$= \lim_{n\to\infty} \exp\Big( \sum_{i=1}^{2^n} \frac{h}{2^n} \frac{\log \mathbb{P}(E_{n,i}| \cap_{j=1}^{i-1} E_{n,j}) - \log 1}{h/(2^n)}\Big)$$

$$= \lim_{n\to\infty} \exp\Big( \sum_{i=1}^{2^n} \frac{h}{2^n} \frac{\int_{\frac{i-1}{2^n}t}^{\frac{i}{2^n}t} Q_{1,s}(x,x)\mathrm{d}s + R_{n,i}}{h/(2^n)}\Big)$$

$$= \exp\Big( \int_0^t Q_{1,s}(x,x)\mathrm{d}s\Big),$$

which completes the proof. $\qquad\square$

## F.2. Proof of Proposition 3

*Proof.* This proof is similar to Proposition 1 and 3 of Wan et al. (2025b). Note that the Poisson process has independent increments. By the conditioning argument (e.g., Theorem 3.7.9 in Durrett, 2019) of the Poisson process $N(t)$, for any $t > 0$, we have

$$\mathbb{P}(N((t,t+h],A) = 1|\mathcal{F}_t) = Mh(1 + \exp(-Mh) - 1)\mathbb{P}(N((t,t+h],A) = 1|N(t+h) - N(t) = 1, \mathcal{F}_t)$$
$$+ \mathbb{P}(N((t,t+h],A) = 1, N(t+h) - N(t) > 1|\mathcal{F}_t)$$
$$= Mh \int_t^{t+h} \sum_{z\in A; z\neq X(t)} \frac{1}{h} \frac{\mathbb{1}(T^X \le t)Q_{1,s}(X(t),z) + \mathbb{1}(T^X > t)Q_{2,s}(X(t),z;X(T^X))}{M}\mathrm{d}s + R_1$$
$$= \int_t^{t+h} \sum_{z\in A; z\neq X(t)} \mathbb{1}(T^X > t)Q_{1,s}(X(t),z) + \mathbb{1}(T^X \le t)Q_{2,s}(X(t),z;X(T^X))\mathrm{d}s + R_1,$$

$$\tag{18}$$

where the remainder $R_1$ satisfying

$$|R_1| \le M^2h^2 + \mathbb{P}(N(t+h) - N(t) > 1)$$
$$\le M^2h^2 + \exp(-Mh)[\exp(Mh) - 1 - Mh]$$
$$\le 2M^2h^2,$$

for sufficiently small $h > 0$. Note that

$$E\Big[\Big\{N((t,t+h],A)\Big\}\mathbb{1}(N((t,t+h],A) \ge 2)\Big|\mathcal{F}_t\Big] \le E\Big[\Big\{N(t+h) - N(t)\Big\}\mathbb{1}(N(t+h) - N(t) \ge 2)\Big]$$
$$= Mh - Mh\exp(-Mh)$$
$$\le M^2h^2.$$

Consequently, we have

$$\mathbb{E}[N((t, t+h], A)|\mathcal{F}_t] = \mathbb{P}(N((t, t+h], A) = 1|\mathcal{F}_t) + E\left[\left\{N((t, t+h], A)\right\}\mathbb{1}(N((t, t+h], A) \geq 2)\Big|\mathcal{F}_t\right]$$

$$= \int_t^{t+h} \sum_{z \in A; z \neq X(t)} \mathbb{1}(T^X > t)Q_{1,s}(X(t), z) + \mathbb{1}(T^X \leq t)Q_{2,s}(X(t), z; X(T^X))\mathrm{d}s + R_2,$$

where $|R_2| \leq 3M^2h^2$ for sufficiently small $h > 0$.

Then, since $X(t)$ has only finite jumps in $[t_1, t_2]$ almost surely, by dominated convergence theorem, we have

$$\mathbb{E}\left[N(t_2, A) - N(t_1, A)\Big|\mathcal{F}_{t_1}\right]$$

$$= \sum_{i=1}^n \mathbb{E}\left[\mathbb{E}[N(t_1 + \frac{i}{n}(t_2 - t_1), A) - N(t_1 + \frac{i-1}{n}(t_2 - t_1), A)|\mathcal{F}_{t_1 + \frac{i-1}{n}(t_2 - t_1)}]\Big|\mathcal{F}_{t_1}\right]$$

$$= \sum_{i=1}^n \mathbb{E}\Bigg\{\int_{t_1 + \frac{i-1}{n}(t_2 - t_1)}^{t_1 + \frac{i}{n}(t_2 - t_1)} \sum_{z \in A; z \neq X(t_1 + \frac{i-1}{n}(t_2 - t_1))} \mathbb{1}(T^X > t_1 + \frac{i-1}{n}(t_2 - t_1))Q_{1,s}(X(t_1 + \frac{i-1}{n}(t_2 - t_1)), z)$$

$$+ \mathbb{1}(T^X \leq t_1 + \frac{i-1}{n}(t_2 - t_1))Q_{2,s}(X(t_1 + \frac{i-1}{n}(t_2 - t_1)), z; X(T^X))\mathrm{d}s\Big|\mathcal{F}_{t_1}\Bigg\} + R_3$$

$$\overset{a.s.}{\to} \mathbb{E}\Big[\int_{t_1}^{t_2} \sum_{z \in A} Q_s(X(s-), z)\mathbb{1}(z \neq X(s-))\mathrm{d}s\Big|\mathcal{F}_{t_1}\Big],$$

as $n \to \infty$, where $|R_3| \leq 3M^2(t_2 - t_1)/n$. Here we use

$$\mathbb{E}\Big[\int_{t_1}^{t_2} \sum_{z \in A; z \neq X(t)} \mathbb{1}(T^X > s)Q_{1,s}(X(s), z) + \mathbb{1}(T^X \leq s)Q_{2,s}(X(s), z; X(T^X))\mathrm{d}s\Big|\mathcal{F}_{t_1}\Big]$$

$$= \mathbb{E}\Big[\int_{t_1}^{t_2} \underbrace{\sum_{z \in A} Q_s(X(s-), z)\mathbb{1}(z \neq X(s-))}_{\text{predictable process}} \mathrm{d}s\Big|\mathcal{F}_{t_1}\Big],$$

since the integrator is Lebesgue measure. Thus, the process

$$\tilde{N}(t, A) = N(t, A) - \int_0^t \sum_{z \in A} Q_s(X(s-), z)\mathbb{1}(z \neq X(s-))\mathrm{d}s$$

is the (martingale-valued) compensated random measure associated with $X(t)$. We are done. $\square$

### F.3. Proof of Theorem 5

*Proof.* By Itô's product formula (e.g., Theorem 4.4.13 in Applebaum, 2009) and Itô's formula (Lemma 1), we have

$$\mathrm{d}[\exp(W(t))] = \exp(W(t-)) \sum_{z \neq X(t-)} \left\{\left[Q_t^{\mathbb{P}}(X(t-), z) - Q_t^{\mathbb{Q}}(X(t-), z)\right]\mathrm{d}t + \left(\frac{Q_t^{\mathbb{Q}}(X(t-), z)}{Q_t^{\mathbb{P}}(X(t-), z)} - 1\right)N(\mathrm{d}t, z)\right\}$$

$$= \exp(W(t-)) \sum_{z \neq X(t-)} \left\{\left(\frac{Q_t^{\mathbb{Q}}(X(t-), z)}{Q_t^{\mathbb{P}}(X(t-), z)} - 1\right)\tilde{N}^{\mathbb{P}}(\mathrm{d}t, z)\right\}, \tag{19}$$

which implies that $\exp(W(t))$ is a $\mathbb{P}$-martingale.

By Equation 19 and Itô's product formula again, we have

$$d[\tilde{N}^{\mathbb{Q}}(t,A)\exp(W(t))] = \tilde{N}^{\mathbb{Q}}(t-,A)\exp(W(t-))\sum_{z\neq X(t-)}\left\{\left(\frac{Q_t^{\mathbb{Q}}(X(t-),z)}{Q_t^{\mathbb{P}}(X(t-),z)}-1\right)\tilde{N}^{\mathbb{P}}(dt,z)\right\} + \exp(W(t-))\tilde{N}^{\mathbb{Q}}(dt,A)$$

$$\underbrace{+\exp(W(t-))\sum_{z\in A\setminus X(t-)}\left\{\left(\frac{Q_t^{\mathbb{Q}}(X(t-),z)}{Q_t^{\mathbb{P}}(X(t-),z)}-1\right)N(dt,z)\right\}}_{\text{quadratic variation}}$$

$$= \tilde{N}^{\mathbb{Q}}(t-,A)\exp(W(t-))\sum_{z\neq X(t-)}\left\{\left(\frac{Q_t^{\mathbb{Q}}(X(t-),z)}{Q_t^{\mathbb{P}}(X(t-),z)}-1\right)\tilde{N}^{\mathbb{P}}(dt,z)\right\} + \exp(W(t-))\tilde{N}^{\mathbb{P}}(dt,A)$$

$$+\exp(W(t-))\sum_{z\in A\setminus X(t-)}\left\{\left(\frac{Q_t^{\mathbb{Q}}(X(t-),z)}{Q_t^{\mathbb{P}}(X(t-),z)}-1\right)\tilde{N}^{\mathbb{P}}(dt,z)\right\}$$

$$= \tilde{N}^{\mathbb{Q}}(t-,A)\exp(W(t-))\sum_{z\neq X(t-)}\left\{\left(\frac{Q_t^{\mathbb{Q}}(X(t-),z)}{Q_t^{\mathbb{P}}(X(t-),z)}-1\right)\tilde{N}^{\mathbb{P}}(dt,z)\right\}$$

$$+\exp(W(t-))\sum_{z\in A\setminus X(t-)}\left\{\left(\frac{Q_t^{\mathbb{Q}}(X(t-),z)}{Q_t^{\mathbb{P}}(X(t-),z)}\right)\tilde{N}^{\mathbb{P}}(dt,z)\right\},$$

which implies that $\tilde{N}^{\mathbb{Q}}(t,A)\exp(W(t))$ is a $\mathbb{P}$-martingale. By Lemma 5.2.11 in (Applebaum, 2009), $\tilde{N}^{\mathbb{Q}}(t,A)$ is a $\mathbb{Q}$-martingale.

Moreover, we have

$$D_{\text{KL}}(\mathbb{Q}_{1:t}||\mathbb{P}_{1:t}) = \mathbb{E}_{\mathbb{P}}\left[\log\left(\frac{d\mathbb{Q}_{1:t}}{d\mathbb{P}_{1:t}}\right)\right]$$

$$= \mathbb{E}_{\mathbb{Q}}\left\{\int_0^t\sum_{z\neq X(s-)}\log\frac{Q_s^{\mathbb{Q}}(X(s-),z)}{Q_s^{\mathbb{P}}(X(s-),z)}N(ds,z) + \int_0^t\sum_{z\neq X(s-)}\left[Q_s^{\mathbb{P}}(X(s-),z)-Q_s^{\mathbb{Q}}(X(s-),z)\right]ds\right\}$$

$$= \mathbb{E}_{\mathbb{Q}}\left\{\int_0^t\sum_{x\neq X(s)}D_F(Q_s^{\mathbb{Q}}(X(s-),z)||Q_s^{\mathbb{P}}(X(s-),z))ds\right\},$$

which completes the proof. $\square$

# G. Proof of Convergence Results

### G.1. Proof of Theorem 1

*Proof.* We divide the proof of Theorem 1 into several steps. We first consider the discretization error in the $k$-th time interval by constructing an auxiliary CTMC with a lower-bounded transition rate on $\mathbb{Z}^{\mathcal{D}}$. Then, we derive the KL divergence of the path measures by Girsanov's theorem. Finally, we can bound the total variation distance by utilizing the chain rule for KL divergence, data-processing inequality and Pinsker's inequality.

Since the proof of the result for tau-leaping and that of the result for Euler solver is similar, we only prove the convergence results for tau-leaping. One can derive the same result for Euler solver by following the same way and using $Q_t^{\text{Euler}}$ instead of $Q_t^{\tau}$. The arguments of this technical proof can also be used in the proof of other samplers.

**Step 0. Introducing additional notations.** Let $X_k(t)$ and $X_k^{\tau}(t)$ be the idealized process and the tau-leaping process with transition rates $Q_t(x,z)$ (equation 3) and $Q_t^{\tau}(x,z)$ in $[t_{k-1},t_k]$ (equation 6), respectively. Define the target set

$$\mathbb{T}_{x_0}(x) = \{z\in\mathbb{Z}^{\mathcal{D}}: x_0+z-x\in\mathcal{S}^{\mathcal{D}}, d^H(x,x_0+z-x)=1\}.$$

Note that in the $k$-th time interval $[t_{k-1},t_k]$, the target set of the idealized process and that of the tau-leaping process are $\mathbb{T}_{X_k(t-)}(X_k(t-))$ and $\mathbb{T}_{X_k^{\tau}(t_{k-1})}(X_k^{\tau}(t-))$, respectively. To construct an auxiliary CTMC, we have to consider the union

of these two sets: $\tilde{\mathbb{T}}_{x_{k-1}}(x) \overset{\triangle}{=} (\mathbb{T}_{x_{k-1}}(x) \cup \mathbb{T}_x(x)) \backslash \{x\}$. Denote the time-Lipschitz constant of the oracle transition rate in the $k$-th time interval as $L_k = \sup_{t \in [t_{k-1}, t_k], d^H(x,z)=1} |\dot{Q}_t(x,z)|$. We denote $M_k = \sup_{t \in [t_{k-1}, t_k]} |\dot{\kappa}_t/(1 - \kappa_t)|$.

**Step 1. Constructing an auxiliary process.** First, we focus on the time interval $[t_{k-1}, t_k]$ conditioning on $X_k(t_{k-1}) = x_{k-1}$. We want to construct a CTMC $X^\epsilon(t)$ with the initial distribution $\delta_{x_{k-1}}$ and rate $Q_t^\epsilon(x,z)$ such that $Q_t^\epsilon(x,z) > \epsilon_k$ for any $z \in \tilde{\mathbb{T}}_{x_{k-1}}(x)$, where $\epsilon_k > 0$ is a small positive constant that we will choose later. Then both $D_{KL}(\mathbb{P}_k || P_k^\epsilon)$ and $D_{KL}(\mathbb{P}_k^\tau || \mathbb{P}_k^\epsilon)$ are finite, where $\mathbb{P}_k$, $\mathbb{P}_k^\tau$ and $\mathbb{P}_k^\epsilon$ are the path measures of the idealized process, the tau-leaping process and the auxiliary process on $[t_{k-1}, t_k]$, respectively. We set $Q_t^\epsilon(x,z) = \{Q_t(x_{k-1}, z - x + x_{k-1}) \vee \epsilon_k\} \delta_{\tilde{\mathbb{T}}_{x_{k-1}}(x)}(z)$ for $z \neq x$ and $Q_t^\epsilon(x,x) = -\sum_{z \in \mathbb{Z}^\mathcal{D} : z \neq x} Q_t^\epsilon(x,z)$. (Here, we set $Q_t(x,z) = 0$ for any $z \notin \mathcal{S}^\mathcal{D}$.)

**Step 2. Bounding KL divergence by Girsanov's theorem.** We consider the exit times

$$T_k = \min \{h > 0 : \Delta X_k(t_{k-1} + h) \neq 0\}.$$

Define the following event

$$\mathcal{E}_k = \left\{ T_k \leq \tau_k \right\},$$

where $\tau_k = t_k - t_{k-1}$. Then by Proposition 2, we have

$$\mathbb{P}_k(\mathcal{E}_k | X_k(t_{k-1}) = x_{k-1}) \leq 1 - \exp\left( -\tau_k \mathcal{D} M_k \right),$$

where we use

$$-Q_t(x,x) \leq \sum_{z \neq x} \sum_{d=1}^{\mathcal{D}} \frac{\dot{\kappa}_t}{1 - \kappa_t} \delta_{x^{\backslash d}}(z^{\backslash d}) p^d(z^d | x) \leq \mathcal{D} M_k,$$

for any $x \in \mathcal{S}^\mathcal{D}$ and $t \in [t_{k-1}, t_k]$.

By Girsanov's theorem (Theorem 5), we can bound the KL divergence $D_{KL}(\mathbb{P}_k || \mathbb{P}_k^\epsilon)$ and $D_{KL}(\mathbb{P}_k^\tau || \mathbb{P}_k^\epsilon)$. Note that

$$
\begin{aligned}
&D_{KL}(\mathbb{P}_k || \mathbb{P}_k^\epsilon) \\
&= \mathbb{E}_{\mathbb{P}_k}\Big[ \int_{t_{k-1}}^{t_k} \Big\{ \sum_{z \in \tilde{\mathbb{T}}_{x_{k-1}}(X(s))} D_F(Q_s(X_k(s), z) || Q_s^\epsilon(X_k(s), z)) \Big\} ds \Big] \\
&= \mathbb{E}_{\mathbb{P}_k}\Big[ \underbrace{\int_{t_{k-1}}^{t_k} \Big\{ \sum_{z \in \tilde{\mathbb{T}}_{x_{k-1}}(X(s))} Q_s(X_k(s), z) \log \frac{Q_s(X_k(s), z)}{Q_s^\epsilon(X_k(s), z)} - Q_s(X_k(s), z) + Q_s^\epsilon(X_k(s), z) \Big\} ds}_{\mathcal{I}_1} \Big].
\end{aligned}
$$

Conditioning on $\mathcal{E}_k$, since $|\tilde{\mathbb{T}}_{x_{k-1}}(x)| \leq 2\mathcal{D}|\mathcal{S}|$, the conditional expectation of quantity $\mathcal{I}_1$ is bounded by

$$\mathbb{E}_{\mathbb{P}_k}(\mathcal{I}_1 | \mathcal{E}_k, X_k(t_{k-1}) = x_{k-1}) \leq 2\tau_k \mathcal{D}|\mathcal{S}|M_k \Big[ 1 \vee \log \frac{M_k}{\epsilon_k} \Big],$$

where we use the first inequality of Lemma 4. Note that when $Q_t(x_{k-1}, z) > \epsilon_k$, $D_F(Q_t(x_{k-1}, z) || Q_t(x_{k-1}, z) \vee \epsilon_k) = 0$; when $Q_t(x_{k-1}, z) \in [0, \epsilon_k]$, $D_F(Q_t(x_{k-1}, z) || Q_t(x_{k-1}, z) \vee \epsilon_k) \leq \epsilon$ for $t \in [t_{k-1}, t_k]$. Conditioning on $\mathcal{E}_k^c$, we have $\tilde{\mathbb{T}}_{x_{k-1}}(X(s)) = \mathbb{T}_{x_{k-1}}(x_{k-1})$ and

$$\mathbb{E}_{\mathbb{P}_k}(\mathcal{I}_1 | \mathcal{E}_k^c, X_k(t_{k-1}) = x_{k-1}) \leq \tau_k \mathcal{D}|\mathcal{S}|\epsilon_k.$$

Consequently, we have

$$
\begin{aligned}
&D_{KL}(\mathbb{P}_k || \mathbb{P}_k^\epsilon) \\
&\leq \mathbb{E}_{\mathbb{P}_k}(\mathcal{I}_1 | \mathcal{E}_k^c, X_k(t_{k-1}) = x_{k-1}) \mathbb{P}_k(\mathcal{E}_k^c | X_k(t_{k-1}) = x_{k-1}) + \mathbb{E}_{\mathbb{P}_k}(\mathcal{I}_1 | \mathcal{E}_k, X_k(t_{k-1}) = x_{k-1}) \mathbb{P}_k(\mathcal{E}_k | X_k(t_{k-1}) = x_{k-1}) \\
&\leq \tau_k \mathcal{D}|\mathcal{S}|\epsilon_k + 2\tau_k^2 \mathcal{D}^2 |\mathcal{S}| M_k^2 \Big[ 1 \vee \log \frac{M_k}{\epsilon_k} \Big],
\end{aligned}
$$

By the second inequality of Lemma 4, we have $D_F(Q_{t_{k-1}}(x_{k-1}, z)||Q_t(x_{k-1}, z) \vee \epsilon_k) \leq \frac{L_k^2 \tau_k^2}{2\epsilon_k} \vee (L_k \tau_k + \epsilon_k)$ for $z \neq x_{k-1}$ and $t \in [t_{k-1}, t_k]$. Then, by the construction of $Q_t^\epsilon$, we have

$$
D_{KL}(\mathbb{P}_k^\tau || \mathbb{P}_k^\epsilon)
$$
$$
= \mathbb{E}_{\mathbb{P}_k^\tau}\Big[ \int_{t_{k-1}}^{t_k} \Big\{ \sum_{z \in \tilde{\mathbb{T}}_{x_{k-1}}(X_k^\tau(s))} D_F(Q_s^\tau(X_k^\tau(s), z)||Q_s^\epsilon(X_k^\tau(s), z)) \Big\} ds \Big]
$$
$$
\leq 2\tau_k \mathcal{D}|\mathcal{S}| \Big[ \frac{L_k^2 \tau_k^2}{2\epsilon_k} \vee (L_k \tau_k + \epsilon_k) \Big].
$$

For estimation error, we have

$$
\mathbb{E}_{\mathbb{D}_n}[D_{KL}(\mathbb{P}_k^\tau || \hat{\mathbb{P}}_k^\tau)] = \mathbb{E}_{\mathbb{D}_n}\Big[ \mathbb{E}_{\mathbb{P}_k^\tau}\Big( \int_{t_{k-1}}^{t_k} \Big\{ \sum_{z \neq X_k^\tau(s)} D_F(Q_s^\tau(X_k^\tau(s), z)||\hat{Q}_s^\tau(X_k^\tau(s), z)) \Big\} ds \Big) \Big].
$$

**Step 3. Choosing $\epsilon_k$ and deriving the final bound.** By Lemma 3, we have $L_k \leq H_k + \mathcal{D}M_k^2$, where $H_k = \sup_{t \in [t_{k-1}, t_k]} |\frac{\ddot{\kappa}_t (1 - \kappa_t) + (\dot{\kappa}_t)^2}{(1 - \kappa_t)^2}|$. Choosing $\epsilon_k = \tau_k(H_k + \mathcal{D}M_k^2)$, since $\mathcal{M}_{x_{k-1}} : \mathbb{Z}^\mathcal{D} \to \mathcal{S}^\mathcal{D}$ is a deterministic function defined in Algorithm 2 for given $x_{k-1}$, then by the chain rule for KL divergence (e.g. Theorem 2.5.3 of Cover, 1999), Pinsker's inequality (e.g. Lemma 15.2 of Wainwright, 2019) and data-processing inequality, we have

$$
\mathbb{E}[TV(p_{1-\delta}, \hat{p}_{1-\delta}^\tau)]
$$
$$
\leq TV(p_{1-\delta}, p_{1-\delta}^\epsilon) + TV(p_{1-\delta}^\epsilon, p_{1-\delta}^\tau) + \mathbb{E}[TV(p_{1-\delta}^\tau, \hat{p}_{1-\delta}^\tau)]
$$
$$
\leq \sqrt{\frac{1}{2} D_{KL}(p_{1-\delta}||p_{1-\delta}^\epsilon)} + \sqrt{\frac{1}{2} D_{KL}(p_{1-\delta}^\tau||p_{1-\delta}^\epsilon)} + \sqrt{\frac{1}{2} \mathbb{E}[D_{KL}(p_{1-\delta}^\tau||\hat{p}_{1-\delta}^\tau)]}
$$
$$
\leq \sqrt{\frac{1}{2}\mathcal{D}|\mathcal{S}| \sum_{k=1}^K \tau_k^2 (H_k + \mathcal{D}M_k^2) + \mathcal{D}^2|\mathcal{S}| \sum_{k=1}^K \tau_k^2 M_k^2 \log(\mathcal{D}M_k \tau_k)^{-1}} + \sqrt{2\mathcal{D}|\mathcal{S}| \sum_{k=1}^K \tau_k^2 (H_k + \mathcal{D}M_k^2) + \varepsilon_{\text{Est}}^\tau}
$$
$$
\leq \sqrt{\frac{5}{2}\mathcal{D}|\mathcal{S}| \sum_{k=1}^K \tau_k^2 H_k} + \sqrt{3\mathcal{D}^2|\mathcal{S}| \sum_{k=1}^K \tau_k^2 M_k^2 \log(\mathcal{D}M_k \tau_k)^{-1} + \varepsilon_{\text{Est}}^\tau},
$$

if $\max_{k \in [K]}\{\tau_k \mathcal{D}M_k\} \leq 1$. This completes the proof.

*Sketch of proof for the Euler sampler.* The proof for the Euler sampler follows the same procedure as above, with $Q_t^\tau$ replaced by $Q_t^{\text{Euler}}$ (equation 7). Specifically, we first construct the same auxiliary rate $Q_t^\epsilon(x, z) = \{Q_t(x_{k-1}, z - x + x_{k-1}) \vee \epsilon_k\}\delta_{\tilde{\mathbb{T}}_{x_{k-1}}(x)}(z)$ in Step 1. Then in Step 2, we apply Girsanov's theorem and the same inequality to bound $D_{KL}(\mathbb{P}_k||\mathbb{P}_k^\epsilon)$ and $D_{KL}(\mathbb{P}_k^{\text{Euler}}||\mathbb{P}_k^\epsilon)$. In Step 3, we choose the same truncation parameter $\epsilon_k = \tau_k(H_k + \mathcal{D}M_k^2)$, which yields the same bound as that for tau-leaping (with $\varepsilon_{\text{Est}}^\tau$ replaced by $\varepsilon_{\text{Est}}^{\text{Euler}}$). $\qquad\square$

### G.2. Proof of Theorem 2

*Proof.* In this proof, we use some notations similar to the proof of Theorem 1. Let $X_k^{\text{Euler}}(t)$ be the process with transition rates $Q_t^{\text{Euler}}(x, z)$ (equation 7) in $[t_{k-1}, t_k]$. We consider the exit times

$$
T_k^{\text{Euler}} = \min\{h > 0 : \Delta X_k^{\text{Euler}}(t_{k-1} + h) \neq 0\}.
$$

Note that $X_k^{\text{Euler}}(t)$ can be constructed by uniformization (Proposition 3) with the Poisson process $N(t)_{t \in [t_{k-1}, t_k]}$ (with rate $-Q_{t_{k-1}}(x_{k-1}, x_{k-1})$, *which dominates the diagonal of the transition rate $Q_t^{\text{Euler}}$ in $[t_{k-1}, t_k]$*). Since $Q_t^{\text{Euler}}$ is time-homogeneous (which means that transition structure is time-independent) in $[t_{k-1}, t_k]$, the first exit time of $X^{\text{Euler}}(t)$ is equal to that of $N(t)$. Define the following events

$$
\mathcal{E}_k^{\text{Euler}} = \left\{ T_k^{\text{Euler}} \leq \tau_k \right\} = \left\{ N(t_k) - N(t_{k-1}) > 0 \right\}.
$$

By Proposition 2, we have

$$\mathbb{P}_k^{\text{Euler}}(\mathcal{E}_k^{\text{Euler}}|X_k^{\text{Euler}}(t_{k-1}) = x_{k-1}) = 1 - \exp\left(\tau_k Q_{t_{k-1}}(x_{k-1}, x_{k-1})\right).$$

By Girsanov's theorem (Theorem 5), we have

$$
\begin{aligned}
&D_{KL}(\mathbb{P}_k^{\text{Euler}}||\mathbb{P}_k) \\
&= \mathbb{E}_{\mathbb{P}_k^{\text{Euler}}}\Big[\int_{t_{k-1}}^{t_k} \Big\{\sum_{z \neq X_k^{\text{Euler}}(s)} D_F(Q_s^{\text{Euler}}(X_k^{\text{Euler}}(s), z)||Q_s(X_k^{\text{Euler}}(s), z))\Big\}\mathrm{d}s\Big] \\
&= \mathbb{E}_{\mathbb{P}_k^{\text{Euler}}}\Big[\int_{t_{k-1}}^{t_k} \underbrace{\Big\{\sum_{z \neq X_k^{\text{Euler}}(s)} Q_{t_{k-1}}^{\text{Euler}}(X_k^{\text{Euler}}(s), z) \log \frac{Q_{t_{k-1}}^{\text{Euler}}(X_k^{\text{Euler}}(s), z)}{Q_s(X_k^{\text{Euler}}(s), z)} - Q_{t_{k-1}}^{\text{Euler}}(X_k^{\text{Euler}}(s), z) + Q_s(X_k^{\text{Euler}}(s), z)\Big\}}_{\mathcal{I}_2}\mathrm{d}s\Big].
\end{aligned}
$$

By conditioning argument (e.g., Theorem 3.7.9 in Durrett, 2019) for the Poisson process $N(t)$, we can derive the lower bound:

$$
\begin{aligned}
&D_{KL}(\mathbb{P}_k^{\text{Euler}}||\mathbb{P}_k) \\
&\geq \mathbb{P}_k^{\text{Euler}}((\mathcal{E}_k^{\text{Euler}})^c|X_k^{\text{Euler}}(t_{k-1}) = x_{k-1})\mathbb{E}_{\mathbb{P}_k^{\text{Euler}}}(\mathcal{I}_2|(\mathcal{E}_k^{\text{Euler}})^c, X_k^{\text{Euler}}(t_{k-1}) = x_{k-1}) \\
&\quad + \mathbb{P}_k^{\text{Euler}}(N(t_k) - N(t_{k-1}) = 1|X_k^{\text{Euler}}(t_{k-1}) = x_{k-1})\mathbb{E}_{\mathbb{P}_k^{\text{Euler}}}(\mathcal{I}_2|N(t_k) - N(t_{k-1}) = 1, X_k^{\text{Euler}}(t_{k-1}) = x_{k-1}) \\
&= \exp\left(\tau_k Q_{t_{k-1}}(x_{k-1}, x_{k-1})\right)\int_{t_{k-1}}^{t_k}\Big\{\sum_{z \neq x_{k-1}} D_F(Q_{t_{k-1}}(x_{k-1}, z)||Q_s(x_{k-1}, z))\Big\}\mathrm{d}s \\
&\quad - \tau_k Q_{t_{k-1}}(x_{k-1}, x_{k-1})\exp\left(\tau_k Q_{t_{k-1}}(x_{k-1}, x_{k-1})\right) \\
&\quad \times \int_{t_{k-1}}^{t_k}\int_{t_{k-1}}^{t_k}\mathbb{1}(t \leq s)\Big(\sum_{x \neq x_{k-1}}\sum_{d=1}^{\mathcal{D}}\delta_{x_{k-1}^{\backslash d}}(x^{\backslash d})\frac{Q_{t_{k-1}}^d(x_{k-1}, x^d)}{-\tau_k Q_{t_{k-1}}(x_{k-1}, x_{k-1})}\Big\{\sum_{d' \neq d}\sum_{z^{d'} \neq x^{d'}} D_F(Q_{t_{k-1}}^{d'}(x_{k-1}, z^{d'})||Q_s^{d'}(x, z^{d'}))\Big\}\Big)\mathrm{d}t\mathrm{d}s \\
&\geq \frac{1}{4}\tau_k\mathcal{D}|\mathcal{S}|L_k' + \frac{1}{32}\mathcal{D}^2|\mathcal{S}|^2\tau_k^2\eta_k,
\end{aligned}
$$

where

$$
\begin{aligned}
L_k' &= \inf_{\mathrm{d}^H(x,z)=1}\int_{t_{k-1}}^{t_k}\Big\{\frac{1}{\tau_k}D_F(Q_{t_{k-1}}(x, z)||Q_s(x, z))\Big\}\mathrm{d}s; \\
\eta_k &= \inf_{\substack{d' \neq d \in [\mathcal{D}], z^{d'} \neq x^{d'}, x^d \neq x_{k-1}^d, \\ \mathrm{d}^H(x, x_{k-1})=1, s \in [t_{k-1}, t_k]}} Q_{t_{k-1}}^d(x_{k-1}, x^d)D_F(Q_{t_{k-1}}^{d'}(x_{k-1}, z^{d'})||Q_s^{d'}(x, z^{d'})).
\end{aligned}
$$

Here, we use $\tau_k Q_{t_{k-1}}(x_{k-1}, x_{k-1}) \geq -\log 2$ ☐

### G.3. Proof of Theorem 3

*Proof.* This proof is similar to the proof of Theorem 1. We will use the notations defined in the proof of Theorem 1.

**Step 1. Constructing an auxiliary process.** Let $X_k^{\text{TC}}(t)$ be the process with rate $Q_t^{\text{TC}}$ in time interval $[t_{k-1}, t_k]$, where $Q_t^{\text{TC}}$ is defined in equation 9. We set $Q_t^\epsilon(x, z) = \frac{\dot{\kappa}_t}{1-\kappa_t}\sum_{d=1}^{\mathcal{D}}\delta_{x^{\backslash d}}(z^{\backslash d})\{[\delta_{x_{k-1}^d}(x^d)p_{1|t}^d(z^d|x_{k-1})] \vee \epsilon_k\}$ for $z \neq x$ and $Q_t^\epsilon(x, x) = -\sum_{z \neq x}Q_t^\epsilon(x, z)$. Define $X_k^\epsilon(t)$ as the CTMC with transition rate $Q_t^\epsilon$ in $[t_{k-1}, t_k]$. Denote $\mathbb{P}_k^{\text{TC}}$ and $\mathbb{P}_k^\epsilon$ as the path measures of $X_k^{\text{TC}}(t)$ and $X_k^\epsilon(t)$, respectively.

**Step 2. Bounding KL divergence by Girsanov's theorem.** By Girsanov's theorem (Theorem 5), we can bound the KL divergence $D_{KL}(\mathbb{P}_k||\mathbb{P}_k^\epsilon)$ and $D_{KL}(\mathbb{P}_k^{\text{TC}}||\mathbb{P}_k^\epsilon)$. By the construction of $Q_t^\epsilon$, using the same arguments as the step 2 in the

proof of Theorem 1, we have

$$
D_{KL}(\mathbb{P}_k||\mathbb{P}_k^\epsilon) = \mathbb{E}_{\mathbb{P}_k}\Big[\int_{t_{k-1}}^{t_k}\Big\{\sum_{z\neq X(s)} D_F(Q_s(X_k(s),z)||Q_s^\epsilon(X_k(s),z))\Big\}\mathrm{d}s\Big]
$$

$$
\leq M_k\mathbb{E}_{\mathbb{P}_k}\Big[\int_{t_{k-1}}^{t_k}\Big\{\sum_{z\neq X(s)}\sum_{d=1}^{\mathcal{D}}\delta_{x^{\backslash d}}(z^{\backslash d})D_F\Big(p_{1|t}^d(z^d|X_k(s))\Big\|\Big[\delta_{x_{k-1}^d}(X_k(s)^d)p_{1|t}^d(z^d|x_{k-1})\Big]\vee\epsilon_k\Big)\Big\}\mathrm{d}s\Big]
$$

$$
\leq \tau_k\mathcal{D}|\mathcal{S}|M_k\epsilon_k + \tau_k^2\mathcal{D}^2|\mathcal{S}|M_k^2\Big[1\vee\log\frac{1}{\epsilon_k}\Big].
$$

Let

$$
L_k^p = \sup_{t\in[t_{k-1},t_k],d\in[\mathcal{D}],x\in\mathcal{S}^{\mathcal{D}},z^d\in\mathcal{S}\backslash\{x^d\}}|\dot{p}_{1|t}^d(z^d|x)|
$$

be the time-Lipschitz constant of the posterior in the $k$-th time interval. Using the same arguments as the step 2 in the proof of Theorem 1, it holds that

$$
D_{KL}(\mathbb{P}_k^{\mathrm{TC}}||\mathbb{P}_k^\epsilon) = \mathbb{E}_{\mathbb{P}_k^{\mathrm{TC}}}\Big[\int_{t_{k-1}}^{t_k}\Big\{\sum_{z\neq X_k^{\mathrm{TC}}(s)} D_F(Q_s^{\mathrm{TC}}(X_k^{\mathrm{TC}}(s),z)||Q_s^\epsilon(X_k^{\mathrm{TC}}(s),z))\Big\}\mathrm{d}s\Big]
$$

$$
\leq \tau_k\mathcal{D}|\mathcal{S}|M_k\Big[\frac{(L_k^p)^2\tau_k^2}{2\epsilon_k}\vee(L_k^p\tau_k+\epsilon_k)\Big].
$$

For estimation error, we have

$$
\mathbb{E}_{\mathbb{D}_n}[D_{KL}(\mathbb{P}_k^{\mathrm{TC}}||\hat{\mathbb{P}}_k^{\mathrm{TC}})] = \mathbb{E}_{\mathbb{D}_n}\Big[\mathbb{E}_{\mathbb{P}_k^{\mathrm{TC}}}\Big(\int_{t_{k-1}}^{t_k}\Big\{\sum_{z\neq X_k^{\mathrm{TC}}(s)} D_F(Q_s^{\mathrm{TC}}(X_k^{\mathrm{TC}}(s),z)||\hat{Q}_s^{\mathrm{TC}}(X_k^{\mathrm{TC}}(s),z))\Big\}\mathrm{d}s\Big)\Big].
$$

**Step 3. Choosing $\epsilon_k$ and deriving the final bound.** By Kolmogorov backward equation Lemma 2, we have $L_k^p \leq \mathcal{D}M_k$. Choosing $\epsilon_k = \tau_k\mathcal{D}M_k$, by the same arguments as the step 3 in the proof of Theorem 1, if $\max_{k\in[K]}\{\tau_k\mathcal{D}M_k\}\leq 1$, we can obtain that

$$
\mathbb{E}[TV(p_{1-\delta},\hat{p}_{1-\delta}^{\mathrm{TC}})]
$$

$$
\leq TV(p_{1-\delta},p_{1-\delta}^\epsilon) + TV(p_{1-\delta}^\epsilon,p_{1-\delta}^{\mathrm{TC}}) + \mathbb{E}[TV(p_{1-\delta}^{\mathrm{TC}},\hat{p}_{1-\delta}^{\mathrm{TC}})]
$$

$$
\leq \sqrt{\frac{1}{2}D_{KL}(p_{1-\delta}||p_{1-\delta}^\epsilon)} + \sqrt{\frac{1}{2}D_{KL}(p_{1-\delta}^{\mathrm{TC}}||p_{1-\delta}^\epsilon)} + \sqrt{\frac{1}{2}\mathbb{E}[D_{KL}(p_{1-\delta}^{\mathrm{TC}}||\hat{p}_{1-\delta}^{\mathrm{TC}})]}
$$

$$
\leq \sqrt{\frac{1}{2}\mathcal{D}^2|\mathcal{S}|\sum_{k=1}^K\tau_k^2M_k^2 + \frac{1}{2}\mathcal{D}^2|\mathcal{S}|\sum_{k=1}^K\tau_k^2M_k^2\log(\tau_k\mathcal{D}M_k)^{-1}} + \sqrt{\mathcal{D}^2|\mathcal{S}|\sum_{k=1}^K\tau_k^2M_k^2 + \varepsilon_{\mathrm{Est}}^{\mathrm{TC}}}
$$

$$
\leq \sqrt{2\mathcal{D}^2|\mathcal{S}|\sum_{k=1}^K\tau_k^2M_k^2\log(\tau_k\mathcal{D}M_k)^{-1} + \varepsilon_{\mathrm{Est}}^{\mathrm{TC}}},
$$

which completes the proof. □

### G.4. Proof of Theorem 4

*Proof.* This proof is similar to the proof of Theorem 1. We will use the notations defined in the proof of Theorem 1.

**Step 1. Constructing an auxiliary process.** Let $X_k^{\mathrm{LC}}(t)$ be the process with transition rate $Q_t^{\mathrm{LC}}$ in time interval $[t_{k-1},t_k]$, where $Q_t^{\mathrm{LC}}$ is the two-stage transition rate with $Q_{1,t}^{\mathrm{LC}}$ and $Q_{2,t}^{\mathrm{LC}}$ defined in equation 11. We set $Q_t^\epsilon(x,z) = \mathbb{1}(T_k \geq$

$t)Q_{1,t}^\epsilon(x,z) + \mathbb{1}(T_k < t)Q_{2,t}^\epsilon(x,z;X(T_k))$ with

$$Q_{1,t}^\epsilon(x,z) = \frac{\dot{\kappa}_t}{1-\kappa_t} \sum_{d=1}^{\mathcal{D}} \delta_{x^{\backslash d}}(z^{\backslash d})\{p_{1|t}^d(z^d|x) \vee \epsilon_k\};$$

$$Q_{2,t}^\epsilon(x,z|X(T_k)) = \frac{\dot{\kappa}_t}{1-\kappa_t} \sum_{d=1}^{\mathcal{D}} \delta_{x^{\backslash d}}(z^{\backslash d})\delta_{X^d(T_k)}(x^d)\{p_{1|t}^d(z^d|X(T_k)) \vee \epsilon_k\},$$

(20)

where $z \neq x$; we let $Q_t^\epsilon(x,x) = -\sum_{z \neq x} Q_t^\epsilon(x,z)$. Define $X_k^\epsilon(t)$ as the CTMC with transition rate $Q_t^\epsilon$ in $[t_{k-1}, t_k]$. Denote $\mathbb{P}_k^{\text{LC}}$ and $\mathbb{P}_k^\epsilon$ as the path measures of $X_k^{\text{LC}}(t)$ and $X_k^\epsilon(t)$, respectively.

**Step 2. Bounding KL divergence by Girsanov's theorem.** By Girsanov's theorem (Theorem 5), we can bound the KL divergence $D_{KL}(\mathbb{P}_k||\mathbb{P}_k^\epsilon)$ and $D_{KL}(\mathbb{P}_k^{\text{LC}}||\mathbb{P}_k^\epsilon)$. Define the following events

$$\mathcal{E}_k^{\text{LC}} = \Big\{X_k(t) \text{ has at least two jumps in } [t_{k-1}, t_k]\Big\}.$$

By uniformization (Proposition 3), $X_k(t)$ can be constructed by a Poisson process $N_k(t)$ with rate $\mathcal{D}M_k$, where we use

$$-Q_t(x,x) \le \sum_{z \neq x} \sum_{d=1}^{\mathcal{D}} \frac{\dot{\kappa}_t}{1-\kappa_t} \delta_{x^{\backslash d}}(z^{\backslash d}) p_{1|t}^d(z^d|x) \le \mathcal{D}M_k,$$

for any $x \in \mathcal{S}^{\mathcal{D}}$ and $t \in [t_{k-1}, t_k]$. When $\tau_k \mathcal{D}M_k \le \log 2$, we have

$$\mathbb{P}_k(\mathcal{E}_k^{\text{LC}}|X_k(t_{k-1}) = x_{k-1}) \le \mathbb{P}(N(t_k) - N(t_{k-1}) \ge 2) \le (\tau_k \mathcal{D}M_k)^2 \exp\Big(-\tau_k \mathcal{D}M_k\Big) \le (\tau_k \mathcal{D}M_k)^2.$$

By Girsanov's theorem (Theorem 5), we have

$$D_{KL}(\mathbb{P}_k||\mathbb{P}_k^\epsilon) = \mathbb{E}_{\mathbb{P}_k}\Big[\underbrace{\int_{t_{k-1}}^{t_k} \Big\{\sum_{z \neq X(s)} D_F(Q_s(X_k(s),z)||Q_s^\epsilon(X_k(s),z))\Big\}\mathrm{d}s}_{\mathcal{I}_3}\Big].$$

Conditioning on $\mathcal{E}_k^{\text{LC}}$, the conditional expectation of quantity $\mathcal{I}_3$ is bounded by

$$\mathbb{E}_{\mathbb{P}_k}(\mathcal{I}_3|\mathcal{E}_k^{\text{LC}}, X_k(t_{k-1}) = x_{k-1}) \le \tau_k \mathcal{D}|\mathcal{S}|M_k\Big[1 \vee \log\frac{1}{\epsilon_k}\Big],$$

where we use the first inequality of Lemma 4. Conditioning on $(\mathcal{E}_k^{\text{LC}})^c$, by the construction of $Q_t^\epsilon$, we have

$$\mathbb{E}_{\mathbb{P}_k}(\mathcal{I}_3|(\mathcal{E}_k^{\text{LC}})^c, X_k(t_{k-1}) = x_{k-1}) \le \tau_k \mathcal{D}|\mathcal{S}|M_k \epsilon_k.$$

Consequently, we have

$$D_{KL}(\mathbb{P}_k||\mathbb{P}_k^\epsilon)$$
$$\le \mathbb{E}_{\mathbb{P}_k}(\mathcal{I}_3|(\mathcal{E}_k^{\text{LC}})^c, X_k(t_{k-1}) = x_{k-1}) + \mathbb{E}_{\mathbb{P}_k}(\mathcal{I}_3|\mathcal{E}_k^{\text{LC}}, X_k(t_{k-1}) = x_{k-1})\mathbb{P}_k(\mathcal{E}_k^{\text{LC}}|X_k(t_{k-1}) = x_{k-1})$$
$$\le \tau_k \mathcal{D}|\mathcal{S}|M_k \epsilon_k + (\tau_k \mathcal{D}M_k)^3 |\mathcal{S}|\Big[1 \vee \log\frac{1}{\epsilon_k}\Big],$$

Let

$$L_k^p = \sup_{t \in [t_{k-1}, t_k], d \in [\mathcal{D}], x \in \mathcal{S}^{\mathcal{D}}, z^d \in \mathcal{S}\backslash\{x^d\}} |\dot{p}_{1|t}^d(z^d|x)|$$

be the time-Lipschitz constant of the posterior in the $k$-th time interval. Using the same arguments as the step 2 in the proof of Theorem 1, it holds that

$$D_{KL}(\mathbb{P}_k^{\text{LC}}||\mathbb{P}_k^\epsilon) = \mathbb{E}_{\mathbb{P}_k^{\text{LC}}}\Big[\int_{t_{k-1}}^{t_k} \Big\{\sum_{z \neq X_k^{\text{LC}}(s)} D_F(Q_s^{\text{LC}}(X_k^{\text{LC}}(s),z)||Q_s^\epsilon(X_k^{\text{LC}}(s),z))\Big\}\mathrm{d}s\Big]$$

$$\le \tau_k \mathcal{D}|\mathcal{S}|M_k\Big[\frac{(L_k^p)^2 \tau_k^2}{2\epsilon_k} \vee (L_k^p \tau_k + \epsilon_k)\Big].$$

(21)

For estimation error, we have

$$\mathbb{E}_{\mathbb{D}_n}[D_{KL}(\mathbb{P}_k^{\text{LC}}||\hat{\mathbb{P}}_k^{\text{LC}})] = \mathbb{E}_{\mathbb{D}_n}\Big[\mathbb{E}_{\mathbb{P}_k^{\text{LC}}}\Big(\int_{t_{k-1}}^{t_k}\Big\{\sum_{z\neq X_k^{\text{LC}}(s)} D_F(Q_s^{\text{LC}}(X_k^{\text{LC}}(s),z)||\hat{Q}_s^{\text{LC}}(X_k^{\text{LC}}(s),z))\Big\}\text{d}s\Big)\Big].$$

**Step 3. Choosing $\epsilon_k$ and deriving the final bound.** Assume that $\max_{k\in[K]}\{\tau_k\mathcal{D}M_k\} \leq \log 2$. Choosing $\epsilon_k = \tau_k^2\mathcal{D}^2M_k^2\{1\vee(L_k^p/(\tau_k\mathcal{D}^2M_k^2))\}$, by the same arguments as the step 3 in the proof of Theorem 1, we can obtain that

$$\mathbb{E}[TV(p_{1-\delta},\hat{p}_{1-\delta}^{\text{LC}})] \leq TV(p_{1-\delta},p_{1-\delta}^\epsilon) + TV(p_{1-\delta}^\epsilon,p_{1-\delta}^{\text{LC}}) + \mathbb{E}[TV(p_{1-\delta}^{\text{LC}},\hat{p}_{1-\delta}^{\text{LC}})]$$

$$\leq \sqrt{\frac{1}{2}D_{KL}(p_{1-\delta}||p_{1-\delta}^\epsilon)} + \sqrt{\frac{1}{2}D_{KL}(p_{1-\delta}^{\text{LC}}||p_{1-\delta}^\epsilon)} + \sqrt{\frac{1}{2}\mathbb{E}[D_{KL}(p_{1-\delta}^{\text{LC}}||\hat{p}_{1-\delta}^{\text{LC}})]}$$

$$\leq \sqrt{\frac{1}{2}\mathcal{D}^3|\mathcal{S}|\sum_{k=1}^K\tau_k^3M_k^3\{1\vee(L_k^p/(\tau_k\mathcal{D}^2M_k^2))\} + \frac{1}{2}\mathcal{D}^3|\mathcal{S}|\sum_{k=1}^K\tau_k^3M_k^3\log(\mathcal{D}M_k\tau_k)^{-2}}$$

$$+ \sqrt{\mathcal{D}^3|\mathcal{S}|\sum_{k=1}^K\tau_k^3M_k^3\{1\vee(L_k^p/(\tau_k\mathcal{D}^2M_k^2))\} + \varepsilon_{\text{Est}}^{\text{LC}}}$$

$$\leq \sqrt{2\mathcal{D}^3|\mathcal{S}|\sum_{k=1}^K\tau_k^3M_k^3\{(L_k^p/(\tau_k\mathcal{D}^2M_k^2))\vee\log(\mathcal{D}M_k\tau_k)^{-2}\} + \varepsilon_{\text{Est}}^{\text{LC}}},$$

which completes the proof. $\qquad\square$

## G.5. Proof of Theorem 6

*Proof.* It suffices to bound equation 21 in the proof of Theorem 4 using convexity of Bregman divergence and choose $\epsilon_k = \min\{\underline{M}_k,\tau_k^2\mathcal{D}^2M_k^2\}$. Then we have

$$D_{KL}(\mathbb{P}_k^{\text{LC}}||\mathbb{P}_k^\epsilon) = \mathbb{E}_{\mathbb{P}_k^{\text{LC}}}\Big[\int_{t_{k-1}}^{t_k}\Big\{\sum_{z\neq X_k^{\text{LC}}(s)} D_F(Q_s^{\text{LC}}(X_k^{\text{LC}}(s),z)||Q_s^\epsilon(X_k^{\text{LC}}(s),z))\Big\}\text{d}s\Big] \leq \tau_k^3\mathcal{D}|\mathcal{S}|M_k\Big[\frac{(L_k^p)^2}{2\underline{M}_k}\Big].$$

Consequently, it holds that

$$\mathbb{E}[TV(p_{1-\delta},\hat{p}_{1-\delta}^{\text{LC}})] \leq TV(p_{1-\delta},p_{1-\delta}^\epsilon) + TV(p_{1-\delta}^\epsilon,p_{1-\delta}^{\text{LC}}) + \mathbb{E}[TV(p_{1-\delta}^{\text{LC}},\hat{p}_{1-\delta}^{\text{LC}})]$$

$$\leq \sqrt{\frac{1}{2}D_{KL}(p_{1-\delta}||p_{1-\delta}^\epsilon)} + \sqrt{\frac{1}{2}D_{KL}(p_{1-\delta}^{\text{LC}}||p_{1-\delta}^\epsilon)} + \sqrt{\frac{1}{2}\mathbb{E}[D_{KL}(p_{1-\delta}^{\text{TC}}||\hat{p}_{1-\delta}^{\text{TC}})]}$$

$$\leq \sqrt{\frac{1}{2}\mathcal{D}^3|\mathcal{S}|\sum_{k=1}^K\tau_k^3M_k^3 + \frac{1}{2}\mathcal{D}^3|\mathcal{S}|\sum_{k=1}^K\tau_k^3M_k^3\log(\mathcal{D}M_k\tau_k)^{-2}} + \sqrt{\mathcal{D}|\mathcal{S}|\sum_{k=1}^K\tau_k^3M_k\Big[\frac{(L_k^p)^2}{4\underline{M}_k}\Big] + \varepsilon_{\text{Est}}^{\text{LC}}}$$

$$\leq \sqrt{2\mathcal{D}^3|\mathcal{S}|\sum_{k=1}^K\tau_k^3M_k^3\underline{M}_k^{-1}\log(\mathcal{D}M_k\tau_k)^{-2} + \varepsilon_{\text{Est}}^{\text{LC}}},$$

where we use $L_k^p \leq M_k\mathcal{D}$. This completes the proof. $\qquad\square$

# H. Additional Experiments

### H.1. Simulation

In this subsection, we present the simulation results in low-dimensional datasets.

**Data distribution.** We consider the data distribution with a blockwise AR(1) structure; that is, we first sample dimension $d=1$ from $X^1(1)\sim\mathcal{U}(\mathcal{S})$ and then we sample dimension $d=2,3$ from

$$X^d(1)|X^{d-1}(1) \sim \begin{cases} 0.9\,\mathcal{U}(X^{d-1}(1)+\{-2,-1,\ldots,2\})+0.1\,\mathcal{U}(\mathcal{S}), & \text{if } X(1)^{d-1}\in[3,|\mathcal{S}|-2] \\ \mathcal{U}(\mathcal{S}), & \text{otherwise} \end{cases},$$

and finally we sample $(X^{3j-2}(1), X^{3j-1}(1), X^{3j}(1))$ from distribution same as $(X^1(1), X^2(1), X^3(1))$ for $j = 2, \ldots, \mathcal{D}/3$ (if $\mathcal{D} > 3$), where $\mathcal{U}(A)$ is the uniform distribution on the set $A \subset \mathcal{S}$. The vocabulary size is 8.

**Experimental setup and implementation details.** We train our logits models using the training objective in Equation 16 for data distributions with dimension $\mathcal{D} \in \{3, 6, 9, 12, 15\}$. All models are MLPs with SiLU activation functions, consisting of four hidden layers with hidden dimension 256. In our simulations, we consider both uniform and masked source distributions, and we train the model for the masked source distribution without time embeddings. We use the Adam optimizer with a learning rate of 1e-3. Each model is trained for 2e5 steps. In the sampling stage, for each sampler (Algorithms 2 to 5), we draw 1e5 samples from the target distribution using a number of steps $K \in [10] \cup \{15, 20, 30, 40, 50, 75, 100\}$. We run each setting 10 times and record the total variation between the empirical data distribution (computed using 1e6 samples) and the estimated distribution (on the first 3 dimensions) and sampling time for each sampler. All experiments are conducted on 4 NVIDIA A100 GPUs.

**Performance comparison of different samplers.** The additional simulation results for masked and uniform source distributions are presented in Figure 5 and Figure 6, respectively. We first compare our corrected samplers with the tau-leaping and Euler samplers. Under the same sampling time, the location-corrected sampler has the best performance among these samplers. For simulations with uniform source distribution, the time-corrected sampler outperforms the tau-leaping and Euler samplers, while in simulations with masked source distribution, it performs comparably to the Euler sampler. Additionally, in simulations with masked source distribution, our location-corrected sampler achieves faster convergence than the tau-leaping, Euler, and time-corrected samplers. Our simulation results also indicate that the time-corrected sampler yields a sampling time comparable to that of the Euler sampler when the number of steps is fixed, and the ratio of the additional computational cost of the location-corrected sampler (relative to the Euler sampler) to the computational cost of the Euler sampler tends to decrease as the number of steps increases. Since we only consider low-dimensional settings in our simulation, the additional computational cost of the location-corrected sampler is negligible when the number of steps is greater than dimension $\mathcal{D}$. We further compare our samplers with the (deterministic midpoint) high-order samplers (Ren et al., 2025b). The corrected samplers consistently outperform the RK2 sampler across all settings. Moreover, under a fixed sampling time, the location-corrected sampler achieves lower total variation than the RK2-Trapezoid sampler for the masked source distribution, and remains competitive with it for the uniform source distribution; this is consistent with our theoretical result that the location-corrected sampler enjoys a lower iteration complexity in the setting with a masked source distribution.

### H.2. Text-to-Image Generation

**Experimental setup and implementation details.** We use the FUDOKI model (Wang et al., 2025) for image generation ($\mathcal{D} = 576, |\mathcal{S}| = 16384$). The metric-induced probability path and conditional transition rate are computed in the same way as in their work. We consider three sampling algorithms (Algorithms 6 to 8) in our experiments. For the corrected samplers, we choose $m = 32$ to compute numerical integrals. For the location-corrected sampler (Algorithm 8), we adaptively choose $j = \mathcal{D}/K$, where $K$ is the number of steps. We consider different settings of the number of steps $K$ and time threshold for the location-corrected sampler $t_\theta$ in experiments. We record the average sampling time (second per image) for each setting and evaluate each sampler on the GenEval benchmark.

**Performance comparison of different samplers.** The experimental results are reported in Table 2, Table 3, and Table 4. From Table 2, we can see that the time-corrected sampler consistently outperforms the Euler sampler using a comparable sampling time, demonstrating improved accuracy without sacrificing efficiency. The location-corrected sampler also achieves comparable performance to the Euler sampler using less sampling time. We also compare our samplers against two (deterministic midpoint) high-order samplers (Ren et al., 2025b). When the sampling time is small (e.g., comparable to Euler with $K = 8$), both RK2 and RK2-Trapezoid perform clearly worse than the time-corrected sampler. From the results in Table 3, we observe that when the number of steps is sufficiently small (say, 4 or 8), reducing the time threshold $t_\theta$ improves accuracy, demonstrating the effectiveness of location correction. In contrast, when the number of steps is relatively large (say, 16 or 32), the overall GenEval score is insensitive to the choice of $t_\theta$. Finally, Table 4 reports the performance of the time-corrected sampler with different numbers of grid points $m$. The results indicate that a moderate number of grid points performs well, while overly large $m$ (64 or 128) may degrade performance.

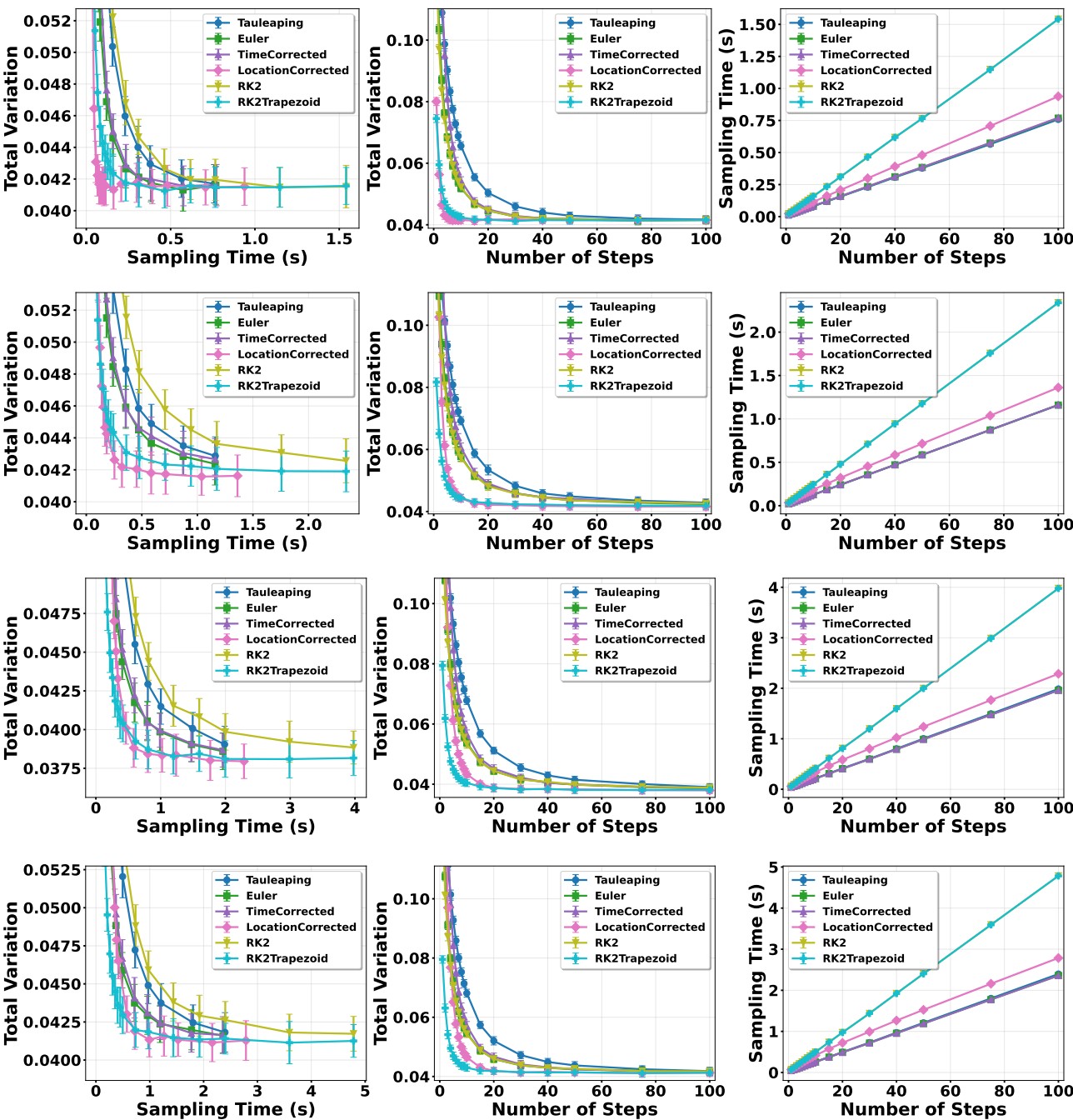

*Figure 5.* Performance comparison ($\mathcal{D} = 3, 6, 12, 15$, from top to bottom) with masked source distribution. (Left) Total variation v.s. sampling time; (Mid) Total variation v.s. number of steps; (Right) Sampling time v.s. number of steps.

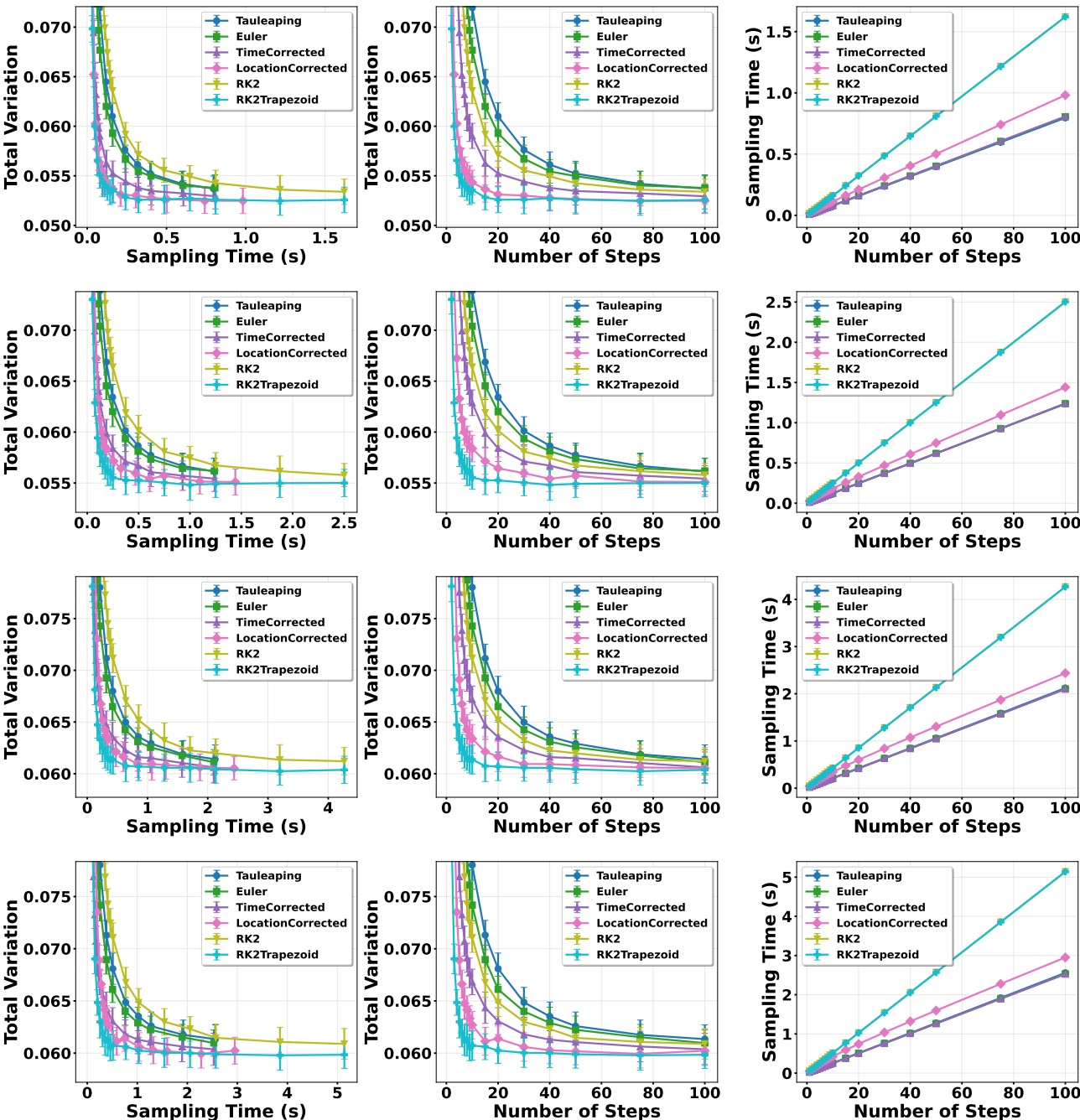

*Figure 6.* Performance comparison ($\mathcal{D} = 3, 6, 12, 15$, from top to bottom) with uniform source distribution. (Left) Total variation v.s. sampling time; (Mid) Total variation v.s. number of steps; (Right) Sampling time v.s. number of steps.

*Table 2.* Performance of different samplers on the GenEval benchmark. We choose time threshold $t_\theta = 0.5$ for location-corrected sampler (Algorithm 8).

| Sampler | # of steps | Time | Single Obj. | Two Obj. | Counting | Colors | Position | Color Attri. | Overall ↑ |
|---|---|---|---|---|---|---|---|---|---|
| Euler | 8 | 3.70482 | 90.00 | 81.82 | 40.00 | 85.11 | 63.00 | 50.00 | 68.321 |
| Time-corrected | 8 | 4.07314 | 96.25 | 79.80 | 42.50 | 93.62 | 65.00 | 56.00 | 72.194 |
| Location-corrected | 4 | 2.50190 | 92.50 | 68.69 | 42.50 | 81.91 | 63.00 | 42.00 | 65.100 |
| RK2 | 4 | 3.23943 | 95.00 | 71.72 | 47.50 | 80.85 | 59.00 | 38.00 | 65.345 |
| RK2-Trapezoid | 4 | 3.23915 | 97.50 | 73.74 | 52.50 | 84.04 | 58.00 | 36.00 | 66.963 |
| Euler | 16 | 7.50117 | 96.25 | 85.86 | 53.75 | 92.55 | 64.00 | 61.00 | 75.569 |
| Time-corrected | 16 | 8.17569 | 95.00 | 84.85 | 48.75 | 89.36 | 70.00 | 66.00 | 75.660 |
| Location-corrected | 8 | 5.58421 | 92.50 | 76.77 | 60.00 | 88.30 | 63.00 | 58.00 | 73.094 |
| RK2 | 8 | 6.99116 | 92.50 | 80.81 | 50.00 | 87.23 | 68.00 | 51.00 | 71.590 |
| RK2-Trapezoid | 8 | 6.99089 | 92.50 | 82.83 | 53.75 | 90.43 | 68.00 | 64.00 | 75.251 |
| Euler | 32 | 14.87659 | 93.75 | 85.86 | 52.50 | 91.49 | 67.00 | 65.00 | 75.933 |
| Time-corrected | 32 | 16.82303 | 95.00 | 78.79 | 53.75 | 91.49 | 70.00 | 72.00 | 76.838 |
| Location-corrected | 16 | 12.06032 | 93.75 | 82.83 | 55.00 | 93.62 | 68.00 | 66.00 | 76.533 |
| RK2 | 16 | 14.45333 | 100.00 | 79.80 | 53.75 | 92.55 | 68.00 | 62.00 | 76.017 |
| RK2-Trapezoid | 16 | 14.45520 | 96.25 | 84.85 | 52.50 | 87.23 | 65.00 | 60.00 | 74.305 |
| Euler | 64 | 30.14049 | 96.25 | 83.84 | 48.75 | 88.30 | 72.00 | 63.00 | 75.356 |
| Time-corrected | 64 | 33.08153 | 91.25 | 86.87 | 51.25 | 91.49 | 66.00 | 70.00 | 76.143 |
| Location-corrected | 32 | 23.95799 | 97.50 | 86.87 | 50.00 | 90.43 | 67.00 | 64.00 | 75.966 |
| RK2 | 32 | 29.87733 | 92.50 | 83.84 | 56.25 | 87.23 | 65.00 | 74.00 | 76.470 |
| RK2-Trapezoid | 32 | 29.88214 | 95.00 | 79.80 | 47.50 | 90.43 | 67.00 | 68.00 | 74.621 |

*Table 3.* Performance of location-corrected sampler with different time threshold $t_\theta$ on the GenEval benchmark.

| Time threshold ($t_\theta$) | # of steps | Time | Single Obj. | Two Obj. | Counting | Colors | Position | Color Attri. | Overall ↑ |
|---|---|---|---|---|---|---|---|---|---|
| 0 | 4 | 3.57322 | 91.25 | 74.75 | 47.50 | 86.17 | 61.00 | 54.00 | 69.111 |
| 0.25 | 4 | 3.06689 | 90.00 | 74.75 | 47.50 | 81.91 | 70.00 | 47.00 | 68.527 |
| 0 | 8 | 7.71165 | 92.50 | 82.83 | 53.75 | 89.36 | 66.00 | 63.00 | 74.573 |
| 0.25 | 8 | 6.67486 | 92.50 | 78.79 | 48.75 | 86.17 | 66.00 | 55.00 | 71.201 |
| 0 | 16 | 16.47159 | 93.75 | 86.87 | 52.50 | 86.17 | 61.00 | 66.00 | 74.381 |
| 0.25 | 16 | 13.77041 | 92.50 | 81.82 | 56.25 | 86.17 | 72.00 | 62.00 | 75.123 |
| 0 | 32 | 32.07903 | 93.75 | 84.85 | 46.25 | 89.36 | 64.00 | 68.00 | 74.368 |
| 0.25 | 32 | 27.70932 | 93.75 | 85.86 | 52.50 | 87.23 | 64.00 | 64.00 | 74.557 |

*Table 4.* Performance of time-corrected sampler with different numbers of grid points $m$ on the GenEval benchmark. The number of sampling steps is $K = 8$.

| $m$ | Time | Single Obj. | Two Obj. | Counting | Colors | Position | Color Attri. | Overall ↑ |
|---|---|---|---|---|---|---|---|---|
| 4 | 3.76407 | 93.75 | 78.79 | 50.00 | 89.36 | 66.00 | 56.00 | 72.317 |
| 8 | 3.79770 | 93.75 | 81.82 | 46.25 | 88.30 | 64.00 | 53.00 | 71.186 |
| 16 | 3.97040 | 93.75 | 85.86 | 47.50 | 87.23 | 67.00 | 54.00 | 72.557 |
| 32 | 4.07314 | 96.25 | 79.80 | 42.50 | 93.62 | 65.00 | 56.00 | 72.194 |
| 64 | 4.47651 | 93.75 | 78.79 | 45.00 | 84.04 | 67.00 | 57.00 | 70.930 |
| 128 | 5.24742 | 93.75 | 79.80 | 41.25 | 86.17 | 58.00 | 53.00 | 68.661 |

