# OpenReview forum: "Corrected Samplers for Discrete Flow Models"
_ICML.cc/2026/Conference — ICML 2026 regular_

### Official Review · Reviewer_TZhV · 2026-02-21

**Soundness:** 3
**Presentation:** 2
**Significance:** 2
**Originality:** 3
**Overall Recommendation:** 5
**Confidence:** 4

**Summary:**

This paper investigates corrected samplers for discrete flow model. Different from vanilla discrete diffusion models that focus on some particular source distribution, DFMs allow any pair of source and target distribution. After providing convergence guarantees for the vanilla tau-leaping and Euler samplers, the paper presents two corrected samplers, the time-corrected and the location-corrected samplers, that show improved rates compared to previous ones. All samplers are evaluated on synthetic data and a text-to-image generation task.

**Compliance With Llm Reviewing Policy:**

Affirmed.

**Final Justification:**

The authors have clearly answered my comments, and I have raised my score.

**Key Questions For Authors:**

Apart from the questions in the weakness section (which are more important), the following questions would be appreciated for my understanding of this paper:

1. How is the proof technique different from that for diffusion models? And how is it related to the special problem of DFM?

2. How is the current method applicable to discrete diffusion sampling?

3. How is the proof of Theorem 1 "similarly" applicable to Euler solver? As far as I know, there is no Girsanov-based analysis for truncated tau-leaping (and thus Euler).

4. Some typos: Equation 2: z and x reversed. Equation 3: why is the second line independent of $p_0$? Lemma 4 header: "Bregman". What is $\delta_{\Tilde{T}}$ on line 1272?

I am happy to raise my score if these questions are properly addressed.

**Limitations:**

yes

**Strengths And Weaknesses:**

Strengths:

1. The paper establishes guarantees for samplers well-known in diffusion models (tau-leaping and Euler)

2. The paper presents two novel samplers that achieve comparable theoretical rates with somewhat improved experimental results

3. Experiment on a real-world task

Weaknesses:

1. On the algorithm design: While it is not clear the quantity to learn in the introduction, it seems from Appendix D and the presented algorithms that the quantity $p_{1|t}$ is learned, with which the rate matrix $Q$ can be derived. This corresponds to the D3PM formulation in the diffusion literature. Then, the reason to employ tau-leaping or Euler style samplers becomes unclear, as one can directly construct the posterior sampling probabilities in a D3PM way, and calculating and using $Q$ seems redundant.

2. Unclarity in theoretical rates: It is not clear how the $\log(1/\delta)$ rate can be achieved. In particular, suppose \tau_k^* M_k = c^* with c^* = poly(1/delta) (using the parameters under Section 5.3), then how to move from poly to log?

3. On numerical simulation: The data points on the real experiment seem noisy.

---

> ### Author Rebuttal · Authors · 2026-03-31
>
> We thank the reviewer for the valuable comments and suggestions. We appreciate the time you spent on the paper. We address your concerns below.  (We answer the reviewer's questions in the same top-to-bottom order as they appear in the reviews.)
>
> Answer for Weaknesses:
>
> **A1**: Thanks for your question. The sampling method in the D3PM paper is to calculate the transition probability from time $t$ to $t-\Delta t$ by using $p(x _{t-\Delta t}=z|x _t=x) \propto \sum _{x _0}p(x _{t-\Delta t}=z|x _t=x,x _0)p(x _0|x _t=x)$ in a discrete-time modeling framework. In parallel, the transition rate in DFM is calculated by using $Q _t(x,z)=\sum _{x _1}Q_t(x,z|x _1)p _t(x _1|x _t=x)$ in a continuous-time framework, which is similar to D3PM. Furthermore, using the transition rate is also convenient for our theoretical analysis.
>
> **A2**: Thank you for this question. You can see that $c^* = \delta^{-1/K} - 1 = e^{\frac{1}{K}\log(\frac{1}{\delta})} - 1 \leq 2(\frac{1}{K}\log(\frac{1}{\delta}))$ provided that $\frac{1}{K}\log(\frac{1}{\delta})\leq \log 2$ (see Line 344), where we use $e^x-1\leq 2x$ for $x\in(0,\log 2)$. In the current work, the quantity of interest is not $\delta$, which is pre-specified at the training stage.
>
>
> **A3**: Thank you for pointing this out. We agree that the real-experiment curves are not perfectly smooth. However, the curves of the two proposed corrected samplers remain above the Euler baseline, which demonstrates the effectiveness of the proposed approach.
>
> Answer for Key questions:
>
> **A4**: Thanks for your great questions. Firstly, our proof technique is similar to that used for diffusion models, and the critical step is to use Girsanov's theorem to bound the KL divergence between two path measures. The main difference between the technical proof of corrected samplers and other existing works is to consider a different event defined in terms of the exit time, which is the core insight of the corrected samplers. Secondly, the transition rate for DFM with a mixture path can be expressed as a time schedule term times a posterior (which covers the masked diffusion; see Section H.4 in [1]). When we parametrize the posterior, the information of the time schedule is known, which motivates our time-corrected samplers. Furthermore, motivated by the randomized midpoint sampler for log-concave sampling, we aim to develop a randomized midpoint sampler for discrete generative models, which can reduce the iteration complexity.
>
> **A5**: Thanks for your question. For the masked diffusion, as we mentioned in the previous answer, where the transition rate can be expressed as a time schedule term times a posterior, we can directly use our corrected samplers.
>
> **A6**: Thank you for this question. As we mentioned in Line 1255, the proof of Theorem 1 for the Euler sampler is the same as the procedure for deriving the bound for tau-leaping, by substituting the tau-leaping rate $Q_t^\tau$ with $Q_t^{\text{Euler}}$. Specifically, we first construct the rate for the auxiliary process $Q _t^\epsilon(x,z)=(Q _t(x _{k-1},z-x+x _{k-1})\vee \epsilon _k)\delta _{d^H(x,z)=1}$. After that, we apply Girsanov's theorem and the same inequality in Step 2 of the proof to bound $D _{KL}(\mathbb{P} _k||\mathbb{P} _k^\epsilon)$ and $D _{KL}(\mathbb{P} _k^\text{Euler}||\mathbb{P} _k^{\epsilon})$. In Step 3, we choose the same truncation parameter $\epsilon_k$ as in the proof for deriving the final bound. We will add the detailed proof (or sketch) for the Euler sampler in the revised version.
>
> **A7**: Thank you for carefully reading.
>
> (1) Equation 2: We will correct the order of $z$ and $x$.
>
> (2) Because the conditional probability path is conditional on $x _1$, the conditional transition rate matrix should satisfy rate-properties conditional on $x _1$, which means that it is only related to the current time and state conditional on $x _1$.
>
> (3) We will fix the Lemma 4 header.
>
> (4) You can find the definition of $\delta _{\tilde{T} _{x _{k-1}}}$ in Line 1265, which is the indicator function of the target set of the auxiliary CTMC $Q _t^\epsilon$.
>
> [1] Shi et al. Simplified and generalized masked diffusion for discrete data. NeurIPS. 2024.

---

> > ### Author Rebuttal · Reviewer_TZhV · 2026-04-01
> >
> > The authors have satisfactorily addressed my concerns. I have raised my score.

---

> > > ### Author Response · Authors · 2026-04-05
> > >
> > > Thank you for your thoughtful reassessment and for recognizing our efforts to clarify and strengthen the paper.

---

### Official Review · Reviewer_vQUK · 2026-03-13

**Soundness:** 3
**Presentation:** 2
**Significance:** 3
**Originality:** 3
**Overall Recommendation:** 5
**Confidence:** 2

**Summary:**

The manuscript derives non-asymptotic TV discretization error bounds for tau-leaping and Euler samplers in DFMs. Standard parallel sampling in discrete spaces induce target mismatches. The authors bound the TV distance via the construction of lower-bounded auxiliary CTMCs and applying a Girsanov-type change-of-measure theorem.
The authors isolate two primary sources of discretization error: the freezing of the time schedule and the freezing of spatial locations. To mitigate these, they propose two algorithms: time-corrected sampler and location-corrected sampler. The location-corrected sampler, formulated as a two-stage jump process, utilizes a randomly sampled exponential exit time (which acts as a stochastic temporal midpoint evaluation), and achieves lower theoretical iteration complexity. Empirical evaluations on a low-dimensional AR(1) process and high-dimensional text-to-image generation (FUDOKI) confirm that the corrected samplers achieve superior TV convergence.

**Compliance With Llm Reviewing Policy:**

Affirmed.

**Key Questions For Authors:**

- Does the magnitude of $\epsilon_{Est}^{Alg}$ empirically dictate the optimal tuning of the time threshold heuristic $t_\theta$ utilized in Algorithm 8? Since the error $\epsilon_{Est}^{Alg}$ is a strictly additive bound, it could be substantial for high dimensional problems.

- How does the iteration complexity of your randomized location-correction algorithm compare to deterministic higher-order tau-leaping like $\theta$-Trapezoidal? Is there merit to using stochastic temporal midpoints over deterministic ones in sparse discrete spaces?

- Is your $\sqrt{|\mathcal{S}|}$ dependence in the non-asymptotic bounds tight? Or is it just an artifact of applying Pinsker's inequality when bounding TV away from KL? Tau-leaping for absorbing diffusions has a linear scaling with $|\mathcal{S}|$.

**Limitations:**

yes

**Strengths And Weaknesses:**

## Strengths

- The analysis is rigoros. Bypassing from the infinite KL problem (caused by target set mismatch) to bounding TV via lower-bounded CTMCs is a robust framework choice. The derivations of the one-step lower-bounds for the Euler sampler effectively motivate the proposed algorithms.

- Presentation is good. The notation is dense but consistent.

- The contribution seems significant. DFMs eliminate the initialization errors of discrete diffusion, but inference inefficiency has bottlenecked their adoption. Both samplers seem to offer significant improvements.

- The approach seems original. Current approaches have focused on accelerating discrete samplers using numerical ODE adaptations, such as the $\theta$-Trapezoidal method, which achieve deterministic second-order convergence. This manuscript provides a stochastic alternative. Moreover, recent work on tau-leaping derives bounds under specific absorbing path assumptions on discrete diffusion, this work tackles the more general conditional path formulations inherent to DFMs.

## Weaknesses
- Theorems 3, 4 require $\tau_k \mathcal{D} M_k \le 1$ or $\le \log 2$. this condition is strictly violated in high-dimensional problems evaluated in the paper (e.g., text-to-image with dimension $\mathcal{D}=576$ at $K \in \{4, 8\}$ steps). As a fix, the authors introduce a fractional order statistic $j$ and a time threshold $t_\theta$ (Algorithm 8) but the theory does not cover this heuristic.

- Table 1 is overly dense; decoupling the complexities explicitly by $\mathcal{D}$, $|\mathcal{S}|$, and $\delta$ would improve clarity.

---

> ### Author Rebuttal · Authors · 2026-03-31
>
> We thank the reviewer for the valuable comments and suggestions. We appreciate the time you spent on the paper. We address your concerns below.
>
> **Q1**: *Theorems 3, 4 require $\tau _k\mathcal{D} M _k \leq 1$ or $\leq \log 2$. This condition is strictly violated in high-dimensional problems evaluated in the paper (e.g., text-to-image with dimension $\mathcal{D}=576$ at $K\in\{4,8\}$ steps). As a fix, the authors introduce a fractional order statistic $j$ and a time threshold $t _\theta$ (Algorithm 8) but the theory does not cover this heuristic.*
>
> **A1**: Thank you for this important observation. Please see Answer 1 in the response to Reviewer nh7K.
>
> **Q2**: *Table 1 is overly dense; decoupling the complexities explicitly by $\mathcal{D}$, $|\mathcal{S}|$, $\delta$ and $K$ would improve clarity.*
>
> **A2**: Thank you for this suggestion. We will restructure Table 1 in the revised version to decouple the complexities more clearly.
>
> Key questions:
>
> **Q3**: *Does the magnitude of $\varepsilon^{\text{Alg}} _{\text{Est}}$ empirically dictate the optimal tuning of the time threshold heuristic $t _\theta$ utilized in Algorithm 8? Since the error $\varepsilon^{\text{Alg}} _{\text{Est}}$ is a strictly additive bound, it could be substantial for high dimensional problems.*
>
> **A3**: Thanks for your question. Firstly, the magnitude of $\epsilon _{\text{Est}}^{\text{Alg}}$ is less related to the discretization error, and the number of steps plays a more important role in the discretization error than the estimation error. As mentioned in Remark 9, the time threshold $t _\theta$ in Algorithm 8 is used for acceleration in the few-step regime. Thus, $\epsilon _\text{Est}^{\text{Alg}}$ is not related to the choice of $t _\theta$. Secondly, the estimation error scales with both dimension $\mathcal{D}$ and sample size, and this error decreases as the sample size increases (empirically one requires more samples in high-dimensional problems). Note that the sample size is less related to the discretization error, and thus we can view the estimation error as a constant for theoretical simplicity.
>
> **Q4**: *How does the iteration complexity of your randomized location-correction algorithm compare to deterministic higher-order tau-leaping like $\theta$-Trapezoidal? Is there merit to using stochastic temporal midpoints over deterministic ones in sparse discrete spaces?*
>
> **A4**: Thanks for the great question. Intuitively, we can imagine that the iteration complexity of our randomized location-corrected sampler is similar to their deterministic midpoint one. However, the proofs of $\theta$-Trapezoidal and $\theta$-RK in that paper mainly rely on Dynkin's formula in an asymptotic setting, and omit other quantities in their bound and the clipping issue. The rigorous proof of the complexity of the deterministic one under our framework needs to be further investigated in future work. For empirical comparison, please see Answer 4 in the response to Reviewer nh7K.
>
> **Q5**: *Is your $\sqrt{|\mathcal{S}|}$ dependence in the non-asymptotic bounds tight? Or is it just an artifact of applying Pinsker's inequality when bounding TV away from KL? Tau-leaping for absorbing diffusions has a linear scaling with $|\mathcal{S}|$.*
>
> **A5**: Thanks for your question. Yes, the $\sqrt{|\mathcal{S}|}$ is indeed a consequence of applying Pinsker's inequality to convert KL to TV. The KL divergence scales linearly with $|\mathcal{S}|$, since the sparse rate matrix has $(|\mathcal{S}|-1)\mathcal{D}+1$ non-trivial entries per row. After applying Pinsker's inequality, it does not change the dependence between $K$ and $|\mathcal{S}|$ when we calculate the iteration complexity (for a fixed estimation error), since they are rooted together. (One might also wonder why the KL is not usually linear in dimension. This is because the magnitude of diagonal entries of the transition rate matrix scales linearly with dimension but is less related to $|\mathcal{S}|$. Thus the dimension is the main quantity of interest in existing works).

---

> > ### Author Rebuttal · Reviewer_vQUK · 2026-04-04
> >
> > I thank the authors for their detailed response. I will maintain my already positive assessment of the work.

---

> > > ### Author Response · Authors · 2026-04-05
> > >
> > > We are very grateful for your kind comments and for your continued positive evaluation of our work.

---

### Official Review · Reviewer_nh7K · 2026-03-13

**Soundness:** 3
**Presentation:** 3
**Significance:** 2
**Originality:** 3
**Overall Recommendation:** 4
**Confidence:** 3

**Summary:**

This paper studies sampling algorithms for Discrete Flow Models (DFMs). The authors establish non-asymptotic error bounds for tau-leaping and Euler samplers without boundedness assumptions on transition rates. Motivated by a one-step lower bound analysis for the Euler sampler, they propose two corrected samplers: (1) a time-corrected sampler that unfreezes the time schedule variable, and (2) a location-corrected sampler that re-evaluates the posterior after the first jump in each interval using a two-stage jump process. Theoretical analysis shows improved iteration complexity for the location-corrected sampler. Experiments on low-dimensional simulations and text-to-image generation demonstrate improved efficiency.

**Compliance With Llm Reviewing Policy:**

Affirmed.

**Final Justification:**

Good paper. I will maintain my positive evaluation.

**Key Questions For Authors:**

Can you reconcile the condition τ_k D M_k ≤ log 2 with your K=4,8 experiments? Do the results hold without this condition, or is there an alternative analysis? For the location-corrected sampler with j=D/K, what is the actual wall-clock time ratio compared to Euler sampler with the same K? Figure 3 suggests location-corrected is slower than time-corrected—does this persist when controlling for total compute? How sensitive are the results to the numerical integration grid size m? Table 3 ablates t_θ but not m. Why does the location-corrected sampler underperform time-corrected at K=16 in Table 2, contrary to the theoretical complexity advantage?

**Limitations:**

yes

**Strengths And Weaknesses:**

## Strengths
The author provides solid theoretical foundation. The non-asymptotic analysis using auxiliary processes and Girsanov-type theorems is technically reasonable. Removing boundedness assumptions on transition rates is a meaningful theoretical advance. The assumption 1 which states estimation error is not strict. Total variation sampler is cleear. This paper propose good algorithmic motivation. The one-step lower bound analysis effectively identifies two distinct error sources-Lipschitz and location-dependence—motivating the two correction strategies. The time-corrected sampler incurs negligible overhead, and the location-corrected sampler offers a complexity improvement from O(D²) to O(D) in ideal cases.

## Weaknesses
1. Theorem 4 requires the condition max_{k∈[K]}{τ_k D M_k} ≤ log 2. For the text-to-image experiments with D=576, K=4, this implies τ_k ≤ log 2 / (576 · M_k) ≈ 0.0012 yet the actual step size is τ_k ≈ 0.25. Tihs point violates the experimental settings. The authors need to explain why the method works empirically despite violating the theorem conditions or provide results under valid regimes.

2. Table 2 shows the time-corrected sampler improves GenEval scores by 0.9 points at K=32 (76.84 vs 75.93), and the location-corrected sampler actually underperforms time-corrected at K=16 (76.53 vs 76.84). Given the additional complexity of location correction, the practical improvement may be limited

3. Computational cost analysis. Section 6.2 and Appendix D.3 propose setting j = D/K for the few-step regime. With D=576, K=4, this yields j=144 parallel updates, each requiring network evaluation. The claim of "almost no additional computational cost" is misleading. Figure 3's time axis obscures this by using adaptive j; a fair comparison would fix computational budget.

4. Missing latest related work comparisons. The paper does not compare against:
- Ren et al. (2025b)'s high-order tau-leaping methods, which also achieve improved convergence rates
- Standard continuous diffusion samplers applied to discretized data, to validate that DFMs with corrected samplers are competitive with established baselines

5. Tightness of bounds. The error bound in Theorem 1 contains terms like τ_k² H_k and τ_k² D² M_k² log(DM_kτ_k)⁻¹. For typical schedules where M_k = κ̇_t/(1-κ_t) → ∞ as t → 1, these bounds blow up. The paper does not discuss whether this fundamental limitation can be removed or how δ should be chosen in practice.

---

> ### Author Rebuttal · Authors · 2026-03-31
>
> We sincerely thank the reviewer for the careful reading, constructive comments, and valuable suggestions. We greatly appreciate your time.
>
> **Q**: *Question about the assumptions and results of real experiments.*
>
> **A**: As noted in the conclusion, existing theory on discretization error mainly focuses on sufficiently large $K$. We discuss the few-step regime in the Appendix and include high-dimensional experiments to extend the theoretical insight empirically. A rigorous theory for the few-step regime is an important future direction.
>
> **Q**: *Question about $K=16, 32$ in table 2.*
>
> **A**: We agree that the empirical gap becomes smaller when $K$ is relatively large. This is consistent with the paper's observation that for $K=16$ or $K=32$, the GenEval score is insensitive to the location-correction threshold (the time-corrected sampler can be viewed as $t_{\theta}=1$), whereas the benefit is more visible in the few-step regime.
>
> **Q**: *Question about computational cost analysis.*
>
> **A**:  Firstly, “Almost no additional computational cost” refers to sufficiently large $K$, since the ratio of the extra function evaluations for location correction to those of Euler goes to zero as $K\to\infty$ (also see simulation study). Secondly, the parameter $j$ for the location-corrected sampler is the order statistic index at the first stage, where we sort the exit times for all tokens (which can be calculated using one function evaluation), and then select the $j$-th smallest time to trigger the update. The parameter $j$ only determines the update time, not the number of function evaluations.
>
> **Q**: *Question about comparing with other samplers.*
>
> **A**: 1. We conduct additional experiments to compare with high-order tau-leaping:
>
> Simulation: $D=9$ (100 repeats), TV (time rel. to Euler), `¹`=best, `²`=second.
>
> ### masked
> | $K$ | 20 | 30 | 50 | 75 |
> |:--|:--:|:--:|:--:|:--:|
> | TC | 0.045635 (1.001) | 0.042078 (0.999) | 0.039913 (1.000) | 0.039021 (1.000) |
> | LC | ¹0.038357 (1.385) | ¹0.037766 (1.306) | ¹0.037505 (1.230) | ¹0.037424 (1.188) |
> | RK2 | 0.045083 (1.986) | 0.041881 (2.001) | 0.039644 (2.013) | 0.038821 (2.020) |
> | Trap | ²0.038567 (1.987) | ²0.037973 (2.000) | ²0.037608 (2.014) | ²0.037562 (2.021) |
>
> ### uniform
> | $K$ | 20 | 30 | 50 | 75 |
> |:--|:--:|:--:|:--:|:--:|
> | TC | 0.060081 (1.000) | 0.059163 (1.000) | 0.058372 (1.000) | 0.057955 (1.000) |
> | LC | ²0.058014 (1.374) | ²0.058008 (1.291) | ¹0.057229 (1.215) | ¹0.057158 (1.174) |
> | RK2 | 0.062449 (2.028) | 0.060728 (2.028) | 0.058995 (2.027) | 0.058508 (2.025) |
> | Trap | ¹0.057129 (2.028) | ¹0.057575 (2.027) | ²0.057385 (2.027) | ²0.057279 (2.026) |
>
> The results indicate that the location-corrected sampler outperforms high-order tau-leaping for masked source and is competitive for uniform source, with less sampling time.
>
> ### Image generation:
> GenEval score (time)
>
> | Method | 4 | 8 | 16 | 32 |
> |:--|:--:|:--:|:--:|:--:|
> | RK2 | 65.345 (3.23943) | 71.590 (6.99116) | 76.017 (14.45333) | 76.470 (29.87732) |
> | Trap | 66.963 (3.23915) | 75.251 (6.99089) | 74.305 (14.44420) | 74.621 (29.88200) |
>
> Combined with the paper's results, the time-corrected sampler has a higher GenEval score than high-order solvers. (For similar sampling time, one can use $K$ for location-corrected sampler and high-order tau-leaping and $2K$ for time-corrected and Euler sampler.)
>
> 2. Our work discusses discrete samplers for discrete generative models on discrete state spaces. A direct comparison with continuous diffusion samplers is beyond the scope of this work.
>
> **Q**: *Question about tightness of bounds and early stopping.*
>
> **A**: Early stopping is necessary for DFM if there is no additional assumption on the source and target distributions. The current paper views $\delta$ as a constant, which is pre-specified at the training stage. If one adds a bounded-rate condition for the interval $[0,1]$, these bounds are finite as $t\to1$. Intuitively, since the time-Lipschitz for the transition rate has the *equality* presented in Line 988, a natural bound for time-Lipschitz would be $H_k+\mathcal{D}M_k^2$. Thus, the upper bound is nearly tight under the current assumption (also note that Theorem 2 for a lower bound, where $L_k^\prime$ is related to the time-Lipschitz regularity).
>
> **Q**: *Question about wall-clock time in real tasks.*
>
> **A**: The wall-clock time is reported in Table 2. In the few-step regime, the expected number of function evaluations for the location-corrected sampler is nearly twice that for the time-corrected sampler, for fixed $K$.
>
> **Q**:  *Question about $m$*:
>
> **A**: We conduct additional experiments for $m$ in the setting of 8 steps:
>
> | $m$ | 4 | 8 | 16 | 32 | 64 | 128 |
> |:--|--:|--:|--:|--:|--:|--:|
> | TC | 72.317 (3.76407) | 71.186 (3.79770) | 72.557 (3.97040) | 72.194 (4.07314) | 70.930 (4.47651) | 68.661 (5.24742) |
>
> The results indicate that a moderate number of grid points performs well, while overly large $m$ (64 or 128) may degrade performance.

---

> > ### Author Rebuttal · Reviewer_nh7K · 2026-04-02
> >
> > Thank you for the detailed feedback. I will maintain my positive evaluation.

---

> > > ### Author Response · Authors · 2026-04-05
> > >
> > > We sincerely appreciate your encouraging feedback and are grateful that our rebuttal has addressed your concerns.

---

### Official Review · Reviewer_ErDd · 2026-03-13

**Soundness:** 3
**Presentation:** 2
**Significance:** 3
**Originality:** 3
**Overall Recommendation:** 4
**Confidence:** 3

**Summary:**

This paper establishes the non-asymptotic discretization error bounds for tau-leaping sampler and Euler solver for discrete flow models, and propose two corrected samplers accordingly, namely time-corrected sampler and location-corrected sampler. They rigorously show that the new samplers have lower discretization error with almost no additional computational cost. Empirical results validate their theoretical findings.

**Compliance With Llm Reviewing Policy:**

Affirmed.

**Final Justification:**

The weaknesses in presentation can be addressed in the final revision. The authors answer address my concern on the hyperparemeters $K$ and $J$. The paper is good from my perspective, but I tend to be conservative as I'm not that farmiliar with the theory of discrete diffusion models.

**Key Questions For Authors:**

Is there a principle for determining the optimal $K$ and $J$ using $J$-stage location-corrected sampler for $K$ time intervals?

**Limitations:**

yes

**Strengths And Weaknesses:**

Strengths:
1. The paper studies an interesting and important problem.
2. The paper seems technically solid, both theoretically and empirically.
3. The proposed methods are insightful.

Weaknesses:
1. Many equations are given without an intuitive explanation, making the paper hard to follow. (eg. eq(10), eq(12))
2. The formatting is poor. (Pg.4 L.195 R, Pg.5 L.244 L, Pg.6 L.283 R)

---

> ### Author Rebuttal · Authors · 2026-03-31
>
> We thank the reviewer for the valuable comments and suggestions. We appreciate the time you spent on the paper. We address your concerns below.
>
> **Q1**: Many equations are given without an intuitive explanation, making the paper hard to follow. (eg. eq(10), eq(12))
>
> **A1**: Thank you for pointing this out. We will add more intuitive explanations for key equations in the revised version. Equation 10 defines the time-corrected transition rate, where the key idea is to only freeze the posterior $p _{1|t _{k-1}}$ while keeping the time schedule $\dot{\kappa} _t/(1-\kappa _t)$ time-varying, so that the exit time distribution can still be computed analytically (see Equation 11). Equation 12 defines the two-stage location-corrected transition rate: the first-stage rate uses the initial posterior at $t _{k-1}$, and after the first jump (at exit time $T _k$), the second-stage rate re-evaluates the posterior at the new location $X(T _k)$ and time $T _k$, thereby correcting the location-dependent error identified in the one-step lower bound. One can also understand the form of these rates by Remark 1 and Equation 6.
>
>
> **Q2**: The formatting is poor. (Pg.4 L.195 R, Pg.5 L.244 L, Pg.6 L.283 R)
>
> **A2**: Thank you for pointing out the formatting issues. We will carefully fix the formatting problems at the indicated locations in the revised manuscript.
>
> Key questions:
>
> **Q3**: Is there a principle for determining the optimal $K$ and $J$ using $J$-stage location-corrected sampler for time intervals?
>
> **A3**: Thanks for your question. Firstly, $K$ is the number of steps, which is related to the sampling time. As the number of steps increases, the discretization error decreases. The optimal scheme for the linear schedule using the multi-stage location-corrected sampler has been discussed in Equation 15 by optimizing the discretization error. Secondly, the choice of $J$ depends on the desired trade-off between accuracy and computational cost. A larger $J$ can potentially reduce the discretization error further, but it also increases the number of function evaluations per step. Also, as we mentioned in Remark 5, when $J$ is large, the time-Lipschitz constant (if the neural network has time-embedding) might dominate the discretization error; thus we recommend $J=2$ in practice.

---

> > ### Author Rebuttal · Reviewer_ErDd · 2026-04-02
> >
> > The authors have satisfactorily addressed my concerns. I will keep my positive score.

---

> > > ### Author Response · Authors · 2026-04-05
> > >
> > > Thank you for your positive acknowledgment and for taking the time to review our responses.

---

### Decision · Program_Chairs · 2026-04-30

**Decision:**

Accept (regular)

**Comment:**

The paper considers samplers for discrete flow matching. First, they provide a strong theoretical analysis for various sampling algorithms with minimal assumptions and utilize these insights to design time-corrected and location-corrected samplers. They rigorously show that these samplers have lower complexity and demonstrate their efficacy empirically.

All the reviewers share a positive opinion about the work and its impact, and the authors were able to answer all the questions during the rebuttal. I recommend acceptance.